# Human whole-exome genotype data for Alzheimer's disease

Yuk Yee Leung[1] ✉, Adam C. Naj[1,2], Yi-Fan Chou[1], Otto Valladares[1], Michael Schmidt[3,4], Kara Hamilton-Nelson[3,4], Nicholas Wheeler[5,6], Honghuang Lin[7], Prabhakaran Gangadharan[1], Liming Qu[1], Kaylyn Clark[1], Amanda B. Kuzma[1], Wan-Ping Lee[1], Laura Cantwell[1], Heather Nicaretta[1], Alzheimer's Disease Sequencing Project*, Jonathan Haines[5,6], Lindsay Farrer[8,9], Sudha Seshadri[10,11], Zoran Brkanac[12], Carlos Cruchaga[13], Margaret Pericak-Vance[3,4], Richard P. Mayeux[14], William S. Bush[5,6], Anita Destefano[9,15], Eden Martin[3,4], Gerard D. Schellenberg[1] & Li-San Wang[1] ✉

The heterogeneity of the whole-exome sequencing (WES) data generation methods present a challenge to a joint analysis. Here we present a bioinformatics strategy for joint-calling 20,504 WES samples collected across nine studies and sequenced using ten capture kits in fourteen sequencing centers in the Alzheimer's Disease Sequencing Project. The joint-genotype called variant-called format (VCF) file contains only positions within the union of capture kits. The VCF was then processed specifically to account for the batch effects arising from the use of different capture kits from different studies. We identified 8.2 million autosomal variants. 96.82% of the variants are high-quality, and are located in 28,579 Ensembl transcripts. 41% of the variants are intronic and 1.8% of the variants are with CADD > 30, indicating they are of high predicted pathogenicity. Here we show our new strategy can generate high-quality data from processing these diversely generated WES samples. The improved ability to combine data sequenced in different batches benefits the whole genomics research community.

The Alzheimer's Disease Sequencing Project (ADSP) was established in 2012 as a key initiative to meet the goals of the National Alzheimer's Project Act (NAPA): to prevent and effectively treat Alzheimer's disease (AD) by 2025. Developed jointly by the National Institute on Aging (NIA) and the National Human Genome Research Institute (NHGRI), the aims of the ADSP are to (1) identify protective genomic variants in older adults at risk for AD; (2) identify new risk variants among AD cases; and (3) examine these factors in multi-ethnic populations to identify therapeutic targets for disease prevention.

The ADSP completed and published the analyses results of the whole-exome sequencing (WES) of 10,836 cases and controls previously released in 2018[1]. The data were generated by three NHGRI-funded Sequencing Centers (Broad Institute, the Baylor College of Medicine's Human Genome Sequencing Center, and Washington University's McDonnell Genome Institute) using Illumina technology and underwent quality control (QC) by the ADSP. The study performed rare variant and gene-based analysis and identified three novel genes (*IGHG3, STAG3,* and *ZNF655*) that were associated with AD[1]. These results remained significant after multiple test corrections and were confirmed/strengthened by replication of four independent datasets. However, the discovery of novel rare variants for AD is still limited by the available sample size.

ADSP has sought to leverage other WES datasets (most of which were generated concurrently with the ADSP's data set in the

A full list of affiliations appears at the end of the paper. *A list of authors and their affiliations appears at the end of the paper.
✉e-mail: yyee@pennmedicine.upenn.edu; lswang@pennmedicine.upenn.edu

collaborative network) to increase the power to detect AD-related rare variants, expanding beyond datasets limited to participants of European (Non-Hispanic White (NHW)) ancestry to include samples from African American (AFA) and Caribbean Hispanic (CHI) ancestry groups (ancestral categorization is discussed further). This collaboration will lead to the generation of the largest yet AD WES data set sharable with the public community. Combining datasets generated in projects that are originally designed for studying AD or other related dementias (ADRD) from different labs across different times (2010-2021) poses new challenges, as each WES data set was generated and processed using different protocols, potentially introducing biases into the combined data set[2–4]. As sequencing cost decreases and technology advances, more sequence data will be available shortly (both whole-genome sequencing [WGS] and WES) and will be generated using different protocols. Due to the desire for joint and meta-analysis, there is a need to process data generated by different platforms efficiently and in a consistent manner.

To ensure all sequence data are processed following best practices with consistency and efficiency, the Genome Center for Alzheimer's Disease (GCAD) in collaboration with the ADSP developed the genomic **v**ariant **c**alling **p**ipeline and data management tool for **A**DSP, VCPA[5]. This is functionally equivalent to the CCDG and TOPMed pipelines[6] and is used for processing WGS data. VCPA has currently been adopted as the official pipeline for processing all ADSP sequence data, as well as data received from the collaborative network, a group of principal investigators (PIs) who have obtained either NIH funding or funding from private foundations involved in sequencing small numbers of AD samples.

Compared to WGS data, WES data focuses on exons, which make up ~1% of the entire genome. The critical challenge unique to WES data harmonization is using different capture kits to sequence samples. The capture kits containing probes that were designed using different reference genomes and versions of gene annotations were made by different vendors over the years.

In this work, we describe the largest publicly available WES data for AD and how it was generated using a new bioinformatics strategy. GCAD built a new computational framework on top of VCPA[5] for processing WES data (VCPA-WES, available at https://bitbucket.org/NIAGADS/vcpa-pipeline/src/master/) by integrating information from multiple capture kits (see Table 1 and Fig. 1 for details) while calling variants at the individual level and joint genotyping across individuals. Corresponding updated QC strategies were specifically developed for this data. Finally, all the individual data and joint-called data were shared with the community via https://dss.niagads.org/ in February 2020.

## Results

### Population substructure analysis

The demographics of this dataset are summarized in Table 2. This dataset contains participants clustering in three major ancestry groups: 13,362 individuals of predominantly European (NHW) ancestry; 4103 individuals of AFA; and 2195 participants of CHI ancestry. Population substructure analysis results are presented in Fig. 2 for **a** NHW, **b** African American, and **c** Caribbean Hispanic. Although these data were generated and made publicly available prior to the report on the treatment of race, ethnicity, and ancestry from the National Academies of Science, Engineering, and Medicine (NASEM)[7], the ancestry classification approach applied here was reviewed and updated to be consistent with the suggested criteria guidelines set forth in the report.

### Characteristics of capture kits

As summarized in Table 1, a total of ten capture kits were used for sequencing 20,504 individuals. Since these files were in different genome builds and file formats, GCAD first standardized them in the same file format and normalized them to the same reference genome build GRCh38. Table 3 contains additional key information about the capture kit contents, including the original number of capture regions per capture kits in the original genome build, the number of lifted-over capture regions per captures (in GRCh38), size of targeted genomic regions, the percentage of capture regions that are within Ensembl v94 exons[8], and the percentage of the Ensembl v94 exons (with flanking bps) that are captured by each of the kits.

There are fundamental differences in the capture designs. The number of capture regions for "Roche_SeqCap_EZ_Exome_Probes_v3.0_Target_Enrichment_Probes" is 1.42-1.98 times higher than the other capture kits. This is caused by the inclusion of miRNA or lncRNA sequences beyond the coding region sequences to the captures by some vendors.

We observed a wide range of differences in terms of the bases covered by each of these capture kits with respect to the human reference genome (from 37 million to 69 million bps). An average of 91.76% of the capture regions per capture kit were annotated as Ensembl exons. In addition, on average 95.22% of these exons were captured by each capture kit.

### Capture kit comparison

We next compared the target region designs among different capture kits. First, the Jaccard similarity measure was calculated on all capture regions at bp level across these kits. To do so, we first broke out all the individual capture kit region files per bp, then we used "1" to denote a bp that was covered by a capture region and "0" vice versa. To measure the overlap or similarity between the data in every pair of capture kit, Jaccard coefficient was calculated. It is defined as the number of bp where both kits are equal to 1, called the 'set intersection', divided by the number of bp where either of the two kits is equal to 1, called the 'set union, displayed in formula: $J(A,B) = \frac{|A \cap B|}{|A \cup B|}$. The Jaccard coefficients for each pair of kits were calculated and visualized in Fig. 3. A value of 1 indicates that the kits are very similar to each other, while a 0 indicates the opposite.

The average of all pair-wise Jaccard similarity scores is 0.586 (SD: 0.038). The two most similar kits are "Illumina_Rapid_Capture_Exome_ICE_kit" and "Nimblegen_VCRome_sequencing_w-Custom_Spike-In_Baits" (Jaccard score = 0.83). Conversely, the two most dissimilar kits are "IDT_xGen_Exome_Whole_Exome_Research_Panel_v1.0_w-Custom_Spike-In_Baits" and "Roche_SeqCap_EZ_Exome_Probes_v3.0_Target_Enrichment_Probes" (Jaccard score = 0.39).

### Data quality—WES compressed reference-oriented alignment maps (CRAMs)

The VCPA-WES pipeline generated all CRAMs without using any capture kit information. Therefore, the differences observed in the CRAM metrics are independent of the capture kits and are primarily due to differences in the sequencing platforms used by the different sequencing centers.

We investigated whether the processed CRAMs were affected by sequencing centers (Fig. 4a) or platforms (Fig. 4b). We compared multiple CRAM metrics generated by VCPA-WES, including (i) percentage of mapped reads; (ii) percentage of duplicated reads; (iii) and percentage of paired reads. We also included the quality of reads (based on a Q score of 30 [Q30]), which is sequencing methods/protocol dependent as a negative control. We compared each of these metrics stratified by sequencing centers or platforms.

From Fig. 4a, we observed that (1) the average mapping rate is 99.6%; (2) 93.4% of samples have <20% duplicated reads; (3) 83.4% of samples have >95% proper pairs; and (4) 97.2% of samples have >80% of alignment with Q30.

We computed the decile values for each metric per sequencing centers/platforms and compared every two distributions using Wilcoxon signed-rank test (pair-wise tests). Most of the

**Table 1 | Summary of the data set with 20,504 WES samples from the nine studies, sequenced using ten different capture kits across 14 sequencing centers**

| Studies | Dataset ID in NIAGADS DSS | Sample count (After QC) | Sequencing platform | Capture kits used | | | | | | | | | |
|---|---|---|---|---|---|---|---|---|---|---|---|---|---|
| | | | | Agilent WES v3 capture region | Agilent WES v4 capture region | Agilent WES v5 capture region | Agilent WES v6 capture region | IDT xGen exome whole-exome research panel v1.0 w-custom spike-in baits | Illumina rapid capture exome (ICE) kit | Nimblegen VCRome sequencing w-custom spike-in baits | Nimblegen VCRome V2.1 | Roche SeqCap EZ exome probes v2.1 target enrichment probes | Roche SeqCap EZ exome probes v3.0 target enrichment probes |
| ADSP Discovery | snd000001 | 10657 | HiSeq 2000 /2500 | | | | | | 4585 (59%) | | 6072 (52%) | | |
| ADGC AA | snd000003 | 3157 | HiSeq 3000 | | | | 3157 (41%) | | | | | | |
| Columbia WHICAP | snd000007 | 3861 | HiSeq 2000 /2500 | | | | | | | | | | 3861 (21%) |
| Miami Families | snd000006 | 108 | HiSeq 2000 /2500 | 61 (74%) | 47 (89%) | | | | | | | | |
| CBD | snd000009 | 346 | HiSeq 2000 /2500 | | | 346 (100%) | | | | | | | |
| PSP | snd000010 | 550 | HiSeq 2000 /2500 | | | | | | | | 550 (100%) | | |
| Knight ADRC | snd000008 | 650 | HiSeq 4000 | | | | | 72 (86%) | | 578 (33%) | | | |
| FASe Families | snd000004 | 1100 | HiSeq 2000 /2500 | | | 714 (63%) | | | | 164 (54%) | 222 (69%) | | |
| Brkanac Families | snd000005 | 75 | HiSeq 2000 /2500 | | | | | | | | | 75 (77%) | |

Shown in this table are the total number of samples, whereas the proportion of cases is shown as percentages in brackets. Footnotes for the sequencing site: the Broad Institute (Broad), Baylor College of Medicine Human Genome Sequencing Center (Baylor), the McDonnell Genome Institute at Washington University (WashU), Institute for Genomic Medicine—Columbia University (IGM-Columbia), Children's Hospital of Philadelphia (CHOP), Functional Genomics Core of the Institute for Diabetes, Obesity and Metabolism (FGC, IDOM), Penn Genome Frontiers Institute—University of Pennsylvania (PGFI), Genentech company, (Genentech), MGI company (MGI), Otogenetics Corporation (Otogenetics).

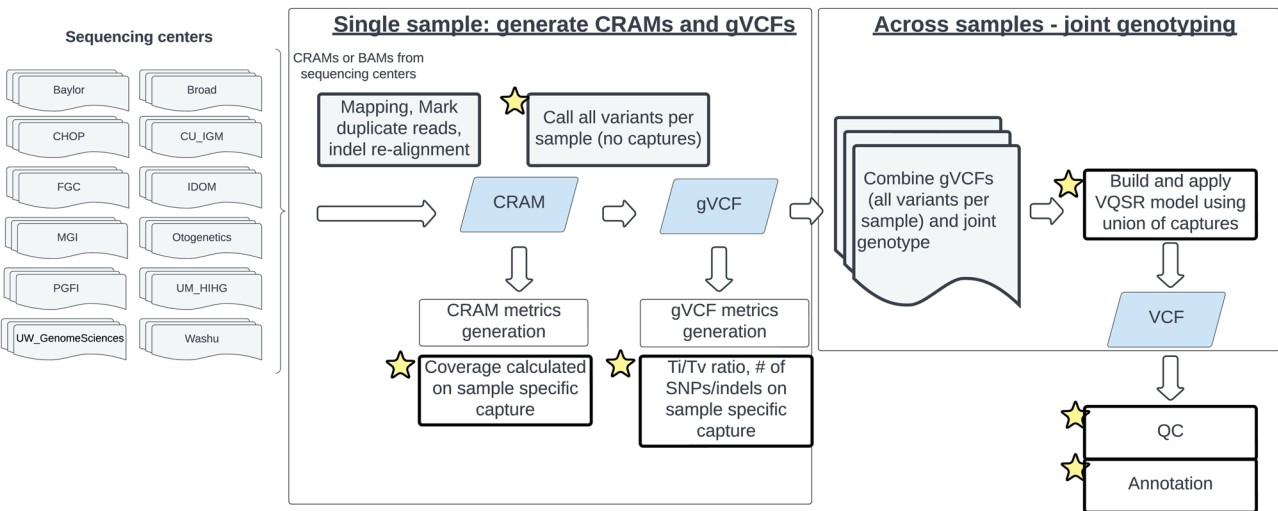

**Fig. 1 | Summary of the VCPA-WES pipeline.** Components with "stars" are modified upon VCPA-WGS pipeline. VCPA-WES specific scripts included: CRAM metrics generation (stage1/wes_depthOfCoverage.sh), gVCF metrics generation (stage2b/wes_no_target_hc_full_bam.sh, stage2b/wes_variantEval.sh), VQSR model generation (stage 3/VQSR_snp_WES.sh, stage 3/VQSR_indel_WES.sh, stage 3/ApplyRecalibration_GATK411_SNP_WES.sh,stage3/ApplyRecalibration_GATK411_indel_WES.sh). All these can be found at https://bitbucket.org/NIAGADS/vcpa-pipeline/src/master/VCPA/.

## Table 2 | Summary of the demographics for each study

| Studies | Race/ethnicity | Cases | Controls | Age Cases | Age Controls | Gender (% of female) Cases | Gender (% of female) Controls | # of APOE e4 alleles (%) Cases 0 | 1 | 2 | Controls 0 | 1 | 2 |
|---|---|---|---|---|---|---|---|---|---|---|---|---|---|
| ADSP discovery | Non-Hispanic White | 5596 | 4279 | 75.6 (8.7) | 86.8 (3.7) | 58 | 59 | 57 | 40 | 3 | 87 | 13 | 0 |
| | Hispanic | 234 | 157 | 75.2 (7.3) | 74.8 (8.4) | 65 | 59 | 59 | 40 | 1 | 61 | 39 | NA |
| | Other/Unknown | 3 | 3 | 76.3 (11.8) | 86.3 (2.1) | 67 | 0 | 67 | 33 | NA | 100 | NA | NA |
| ADGC AA | Non-Hispanic White | 1 | 0 | 58.0 (0) | NA | 0 | NA | NA | 100 | NA | NA | NA | NA |
| | Hispanic | 2 | 6 | 81.0 (0) | 74.0 (12.0) | 100 | 100 | 50 | 50 | NA | 67 | 33 | NA |
| | Black or African American | 1283 | 1634 | 74.5 (8.0) | 72.9 (8.2) | 70 | 74 | 30 | 49 | 15 | 62 | 35 | 3 |
| Columbia WHICAP | Non-Hispanic White | 83 | 800 | 85.4 (5.1) | 80.5 (6.6) | 66 | 58 | 80 | 18 | 2 | 77 | 21 | 2 |
| | Hispanic | 511 | 1257 | 84.0 (5.5) | 80.6 (6.3) | 75 | 70 | 69 | 29 | 2 | 79 | 20 | 1 |
| | Black or African American | 218 | 939 | 84.0 (5.7) | 80.1 (6.6) | 76 | 69 | 61 | 33 | 5 | 67 | 31 | 2 |
| Miami families | Non-Hispanic White | 86 | 18 | 74.2 (7.6) | 76.6 (6.9) | 66 | 39 | 44 | 53 | 2 | 78 | 22 | NA |
| CBD | Non-Hispanic White | 335 | 0 | 63.5 (8.4) | NA | 45 | NA | NA | NA | NA | NA | NA | NA |
| PSP | Non-Hispanic White | 550 | 0 | 69.2 (8.4) | NA | 45 | NA | NA | NA | NA | NA | NA | NA |
| Knight ADRC | Non-Hispanic White | 224 | 338 | 68.3 (8.8) | 71.3 (8.9) | 42 | 59 | 48 | 42 | 9 | 67 | 29 | 3 |
| | Hispanic | 0 | 3 | NA | 78.0 (19.1) | NA | 100 | NA | NA | NA | 100 | NA | NA |
| | Black or African American | 26 | 3 | 64.0 (9.2) | 74.3 (5.5) | 65 | 33 | 19 | 50 | 23 | 67 | 33 | NA |
| | Other/Unknown | 3 | 2 | 76.0 (14.1) | 76.0 (9.9) | 100 | 0 | 33 | NA | 67 | 50 | 50 | NA |
| FASe families | Non-Hispanic White | 731 | 274 | 78.5 (7.2) | 78.8 (7.5) | 63 | 57 | 25 | 57 | 18 | 52 | 44 | 4 |
| | Hispanic | 7 | 2 | 79.4 (3.3) | 75.0 (4.2) | 71 | 50 | 71 | 14 | 14 | 50 | 50 | NA |
| | Other/Unknown | 2 | 2 | 72.5 (3.5) | 68.0 (4.2) | 100 | 0 | 50 | 50 | NA | NA | 100 | NA |
| Brkanac families | Non-Hispanic White | 44 | 0 | 74.9 (7.6) | NA | 61 | NA | 30 | 39 | 25 | NA | NA | NA |
| | Hispanic | 16 | 0 | 68.1 (11.9) | NA | 56 | NA | 63 | 31 | 6 | NA | NA | NA |
| Total | | 9955 | 9717 | 75.8 (8.9) | 81.5 (8.1) | 60 | 64 | 47 | 38 | 6 | 77 | 22 | 1 |

Listed in this table are the total number of samples. Duplicate samples from the same subject for platform comparison are counted multiple times.

comparison results were not statistically significant (Bonferroni corrected $p < 0.05$).
- Stratified by sequencing centers (Fig. 4a), 11%, 3%, 8%, and 76% per "percentage of mapped reads", "percentage of duplicated reads", "percentage of paired reads" and "quality of reads (based

on a Q score of 30 [Q30])" respectively were statistically significant (Bonferroni corrected $p < 0.05$) (Supplementary Tables 1–4).
- Stratified by sequencing platforms (Fig. 4b), 33%, 0%, 0%, and 100% per "percentage of mapped reads", "percentage of

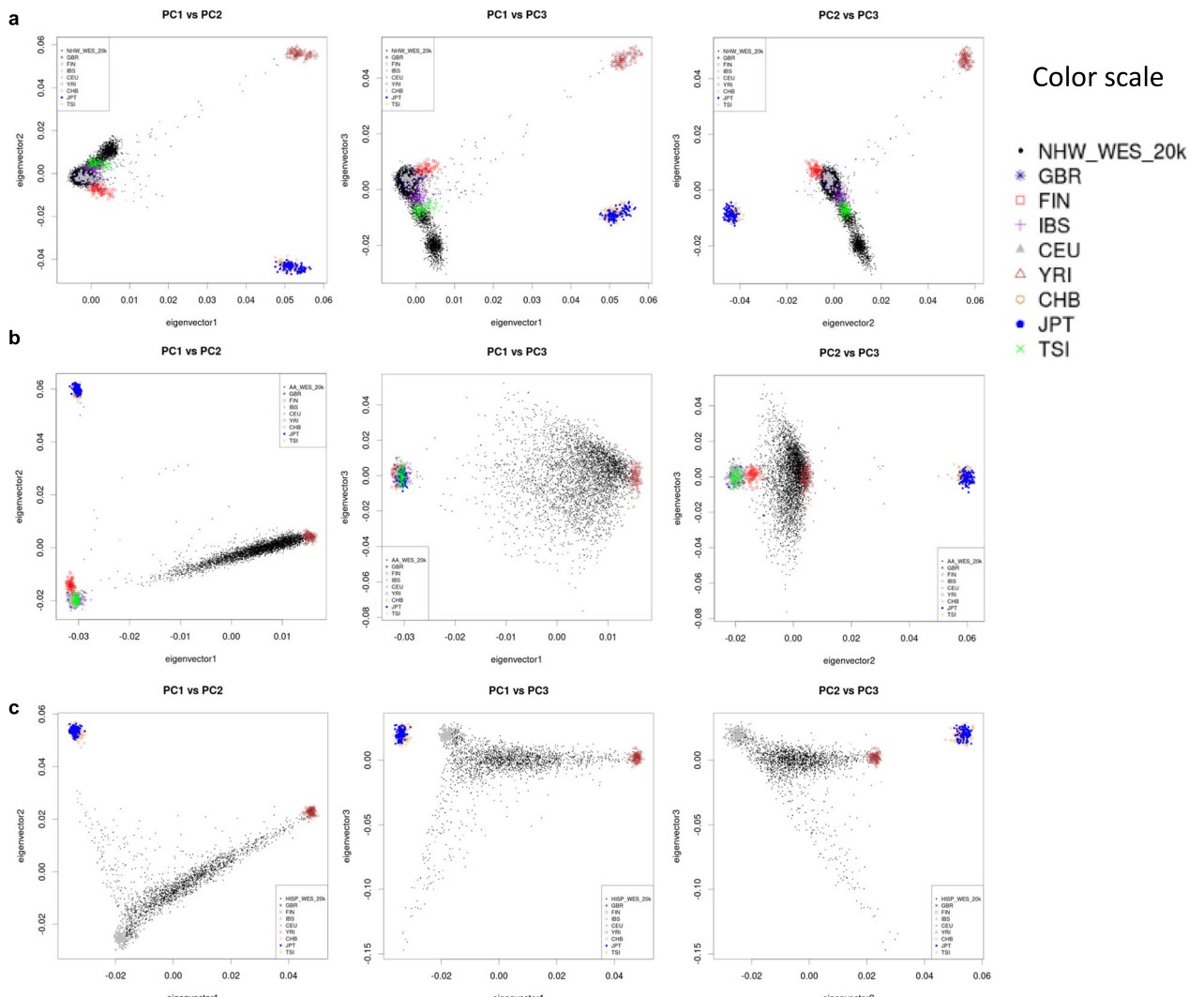

**Fig. 2 | Population substructure analysis results of our dataset.** Plots from principal components analysis showing principal component (PC) 1 vs. PC2, PC2 vs. PC3, and PC1 vs. PC3 for sets of samples initially clustered on self-reported race/ethnicity (samples shown in black dots) with respect to 1kG reference populations (all other symbols). **a** Individual self-reporting as non-Hispanic White and clustering within 3 SD of EUR sample populations were assigned the ancestry label "Non-Hispanic White" (NHW). This plot includes 32 individuals excluded as outliers. **b** Individuals self-reporting as non-Hispanic Black and clustering within 3 SD of EUR sample populations or distributed between the populations were assigned the ancestry label "African American" (AFA). This plot includes 29 individuals excluded as outliers. **c** Individuals clustering within 3 SD of EUR and AFR sample populations and Latin American sample populations groups in the 1000 Genomes/Human Genome Diversity Project collection and between those sample population groups were assigned the ancestry label "Caribbean Hispanic" (CHI), which was also reflective of the geographic sampling of samples in the source datasets. No subjects initially classified as CHI were excluded.

duplicated reads", "percentage of paired reads" and "quality of reads (based on a Q score of 30 [Q30])" respectively were statistically significant (Bonferroni corrected $p < 0.05$) (Supplementary Tables 5–8).

Overall, sequencing centers/platforms have a bigger effect on "Quality of reads, Q30", as this metric reflects the quality of the reads off the sequencers, which is directly dependent on the sequencing methods/protocols. The drastic differences observed indicated that there are indeed differences at the sequencing methods level, yet after processing all the data using the same bioinformatics approach we proposed in this paper (VCPA-WES), the variabilities of data contributed by sequencing centers and platforms have much been reduced (as seen from the metrics at the mapping, duplicated or paired reads level).

While there is some variability in these metrics, we do not observe systematic bias as to which sequencing center performed best/worst in all areas as compared to the others (Fig. 4a, b).

Next, we sought to compare the 20× coverage (i.e., percentage of bps with 20 reads or more within the sample-specific capture regions) across all samples (Fig. 5). 20× coverage was chosen as it was the minimum coverage required to successfully genotype 95% of heterozygous SNPs in an analyses[4,9–13]. For about 95% of the CRAMs, we observed that >80% of reads were located within the capture region at 20× coverage. This metric does have sequencing center- or sequencer-specific effects (Supplementary Tables 9–10). This is as expected as the 20× coverage was calculated on the capture regions per sample, of which the differences are due to the selection of various sequencing methods/protocols. On average, 20× coverage is lower for samples sequenced using the Illumina 2000/2500 platform.

Since there is a high variability of capture kits, we also selected 100 samples based on different studies-sequencing_center-capture combinations to calculate the 20× coverage values just in the coding regions (exons) instead of everything in the capture kit. Results were shown in a scatter plot (Supplementary Fig. 1). We observed that the 20× coverage at the capture regions ($88.5 \pm 7.8$) is similar to that of the

**Table 3 | Characterization of genomic regions (2nd to 3rd columns in GRCh38) by each capture kit**

| Capture kit info | Number of original capture regions (various genome builds) | Number of lifted-over capture regions (GRCh38) | Number of bp | % of capture regions that are Ensembl v94 exons | % of Ensembl v94 exons (with flanking 7bp) that are captured |
| --- | --- | --- | --- | --- | --- |
| Agilent WES v3 capture region | 206,773 | 206,445 | 52,941,356 | 92.54 | 95.13 |
| Agilent WES v4 capture region | 185,638 | 185,600 | 53,791,945 | 92.11 | 94.14 |
| Agilent WES v5 capture region | 230,418 | 230,348 | 53,309,273 | 89.97 | 96.27 |
| Agilent WES v6 capture region | 232,406 | 231,980 | 61,549,327 | 87.78 | 95.88 |
| IDT xGen Exome Whole-Exome Research Panel v1.0 w-Custom Spike-In Baits | 203,403 | 203,310 | 39,968,963 | 94.05 | 95.20 |
| Illumina rapid capture exome (ICE) kit | 217,234 | 216,807 | 40,653,755 | 91.17 | 98.03 |
| Nimblegen VCRome sequencing w-Custom Spike-In Baits | 244,944 | 244,828 | 41,708,836 | 96.26 | 93.63 |
| Roche Nimblegen VCRome V2.1 | 197,542 | 195,167 | 37,951,907 | 96.05 | 94.39 |
| Roche SeqCap EZ Exome Probes v2.0 Target Enrichment Probes | 259,651 | 244,490 | 47,294,869 | 94.68 | 92.34 |
| Roche SeqCap EZ Exome Probes v3.0 Target Enrichment Probes | 368,146 | 367,730 | 69,167,416 | 83.01 | 97.14 |

bp basepairs. First column shows the original number of capture regions per capture kit in the original genome build. The second column shows the number of lifted-over capture regions per capture kit (in GRCh38), while the rest of the columns show the size of targeted genomic regions, the percentage of capture regions that are within Ensembl v94 exons, and the percentage of the Ensembl v94 exons (with flanking bps) that are captured by each of the kits.

coding regions (88.2 ± 2.5) across samples, yet the 20× coverage values in the coding regions are more uniform across samples.

### Data quality−variants

Next, we examined the variant-level data quality. GATK outputs VQSR scores. Overall, 96.83% of the variants (>7.3 million SNVs and 0.61 million indels) were labeled PASS by the model.

Besides using the GATK VQSR indicator to specify the quality of a variant, the ADSP/GCAD QC pipeline[14] outputs a series of quality metrics to determine whether the variant is within capture regions, the call rate, depth, and Ti/Tv ratio after QC. Figure 6 shows the Ti/Tv ratio on the exonic variants (colored by study) before and after QC (Before QC on the x axis, After QC on the y-axis). Before QC, the average Ti/Tv ratio is 2.53. After QC, the Ti/Tv ratio on exonic regions in our studies is around 3.03. This post-QC Ti/Tv ratio is similar to reports in previous findings[10].

The QC protocol enables us to look for variants found across studies as well as those that are study-specific. On average, 97.26% of variants have a GATK PASS across study-capture combinations. 91.45% of variants within the designed capture kit per each study-capture combination are labeled as good quality.

We then sought to compare the quality of variants pre- and post-QC at different aspects. First, we compared the rates of synonymous variants between cases and controls exome-wide first at the capture level (Supplementary Table 11), then at the gene level (Supplementary Fig. 2). We selected the QC subsets from studies of the biggest sample size (ADSP_Discovery, ADGC_AA, and Columbia_WHICAP) for this analysis. Overall, the average ALT allele frequencies across synonymous variants are similar between cases and controls for each QC subset. While most QC subsets had average ALT allele frequencies of 0.01−0.03, a higher frequency was observed in the ADGC African American subset (ADGC_AA_Agilent_WES_v6_capture_region) of 0.046, suggesting a higher frequency of polymorphic synonymous variants among cases relative to controls in this subset.

We also compared the REF/REF, REF/ALT, and ALT/ALT genotype counts summed across coding variants before and after QC for each QC subset (Supplementary Table 12). Variants that were monomorphic or off-capture were excluded from both pre- and post-QC counts. We have also shown in (Supplementary Fig. 3) the "Ratio of Post-QC to Pre-QC Genotype Counts" across QC subsets for all cohort-capture combinations. The ratio of all genotypes is fairly consistent across different QC subsets (0.922-0.956).

Lastly, we compared the genotype call rates across QC subsets pre- and post-QC (Supplementary Fig. 4). In this plot, we showed the percentage of variants by deciles of call rate for each QC subset. Across all but 3 QC subsets (a total 14), we showed the call rates increased modestly once low-quality genotypes were excluded.

### Genotype Concordance between two different callers on a set of overlapping individuals

The ADSP-Discovery data set, comprised of 10,786 individuals, was sequenced and processed by three sequencing centers: Broad Institute, Baylor College of Medicine's Human Genome Sequencing Center, and Washington University's McDonnell Genome Institute. Genotypes for bi-allelic SNVs and indels were called using ATLAS2 on hg19/GRCh37[1]. To evaluate the genotype quality on our 20k WES call set, which was generated using a novel approach in which no capture regions were used for individual sample calling, we examined the overall concordance, by sample and by variant, of genotypes called differently on the 10,786 samples that were present in both the ADSP-Discovery data set and the current data set. 1,407,006 variants were called in both sets, comprising 15,175,966,716 genotypes. Overall concordance was 99.43%. There were five samples with a genotype concordance <95%. Three samples had extremely low concordance (8.21%, 10.44%, and 23.44%), reflecting low DNA concentration

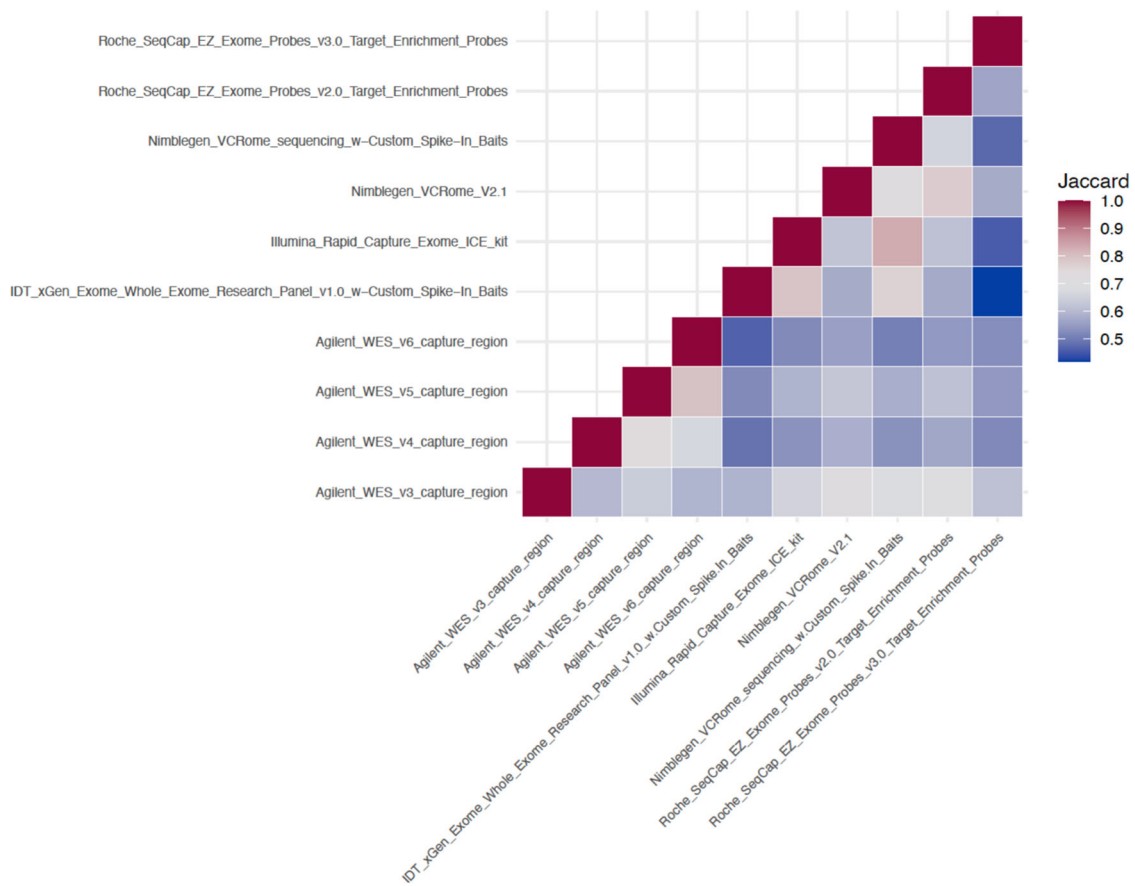

**Fig. 3 | Jaccard similarity measure of the capture kits.** Jaccard similarity measure was calculated on all capture regions (labeled on both the *x* axis and *y* axis) at basepair level across these kits. A value of 1 (dark red) in this figure indicates that the kits are very similar to each other, while a 0 (blue) indicates the opposite.

samples, and the other two samples had 86.4% and 90.5% concordance. We also examined variant-level genotype concordance relative to capture kit coverage. The majority (69.3%) of variants were covered by all ten capture kits, 18.1% by nine, 7% by eight, and 4% by seven capture kits. These patterns may be due to differences in sequence read coverage from the various capture kits combined with joint-calling approaches that leverage information across samples, but this trend only holds for a very small percentage of all called variants.

**Annotation results**

Using the ADSP annotation pipeline[15], we found that the 8.16 million variants are located in 28,579 transcripts based on Ensembl annotations[15,16]. Every variant is annotated based on the most damaging VEP predicted consequence (see "Annotation protocol for WES samples" section). Supplementary Fig. 5a shows the top ten most damaging consequence categories. In summary, 41% of the variants are intronic, 15% are missense variants, followed by upstream/downstream gene variants (each 9%), synonymous variants (8%), and 3'UTR variants (7%). Supplementary Fig. 5b shows the proportion of the CADD score[17] that is PHRED-like scores ranging from 1 to 99, based on the rank of each variant relative to all possible 8.6 billion substitutions in the human reference genome. The mean CADD score of all variants is 9.26, while the median value is 6. Meanwhile, 15.5% of the variants have a CADD score >20, meaning that these variants are among the top 1% of deleterious variants in the human genome. In addition, 1.8% of the WES variants are among the top 0.1% of deleterious variants in the human genome (CADD > 30).

Next, we compared the percentage of high-impact coding variants found in this WES dataset against the gnomADv2.1 public resource[18]. The gnomADv2.1 (GRCh38) contains 17.2 million variants in 125,748 WES samples as compared to 8.2 million variants in 20,504 WES samples in the current dataset. 10.6 million (62%) and 4.0 million (49%) are coding variants in the gnomADv2.1 and this WES data, respectively. Next, we compared the percentage of variants that are annotated as 'high impact' in each set. There are 711,024 (6.7% of coding) and 220,987 (5.5% of coding) high-impact coding variants in the gnomADv2.1 and the current dataset, respectively.

**Data sharing—NIAGADS data sharing service (DSS)**

The National Institute on Aging Genetics of Alzheimer's Disease Data Storage Site (NIAGADS) is a national data repository that facilitates access to genetic data by qualified investigators for the study of the genetics of early-onset/late-onset Alzheimer's Disease and Alzheimer's Disease Related Dementias (ADRD). Collaborations with large consortia such as the Alzheimer's Disease Genetics Consortium (ADGC), Cohorts for Heart and Aging Research in Genomic Epidemiology (CHARGE) Consortium, and the ADSP, a main mission of NIAGADS is to manage large AD genetic datasets that can be easily accessed by the research community.

The NIAGADS Data Sharing Service (DSS) released the CRAMs (compressed version of BAM files), gVCFs generated by GATK4.1.1, and QC-ed pVCFs of the abovementioned ADSP WES data set in September 2020 (NG00067.v3), together with the capture kits, pedigree structures for family studies and phenotypes that were harmonized according to ADSP protocols. Qualified investigators can access these data with a submission request and approval from the NIAGADS Data Access Committee managed by independent NIH program officers. Data can be downloaded through the DSS portal. More information about the data set can be found on the data set page, NG00067). https://dss.niagads.org/datasets/ng00067/. See the Application

**a**   sequencing centers

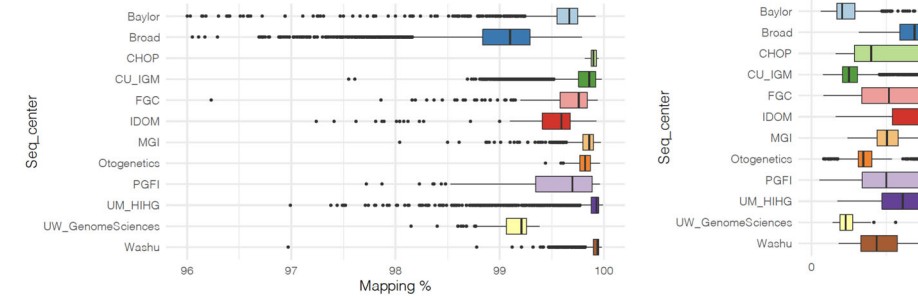

**i) Percentage of mapped reads**

**ii) Percentage of duplicated reads**

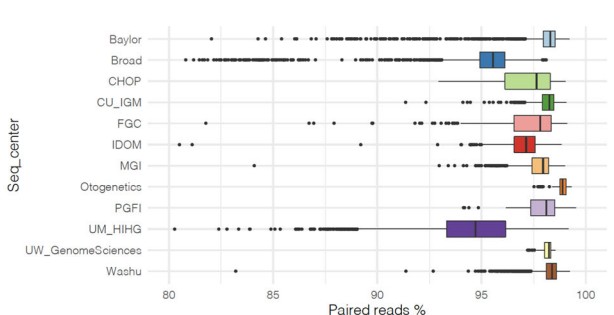

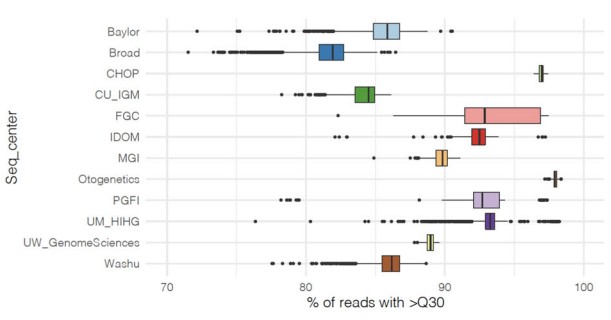

**iii) Percentage of paired reads**

**iv) Quality of reads (based on Q30 score)**

**b**   sequencing platforms

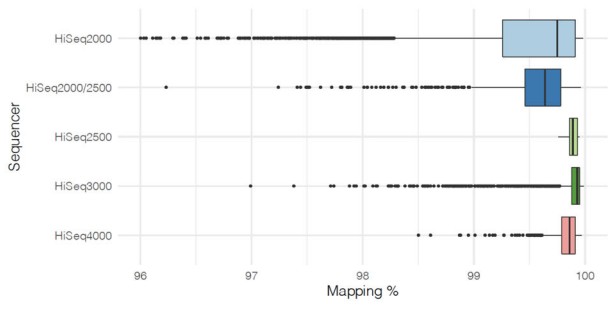

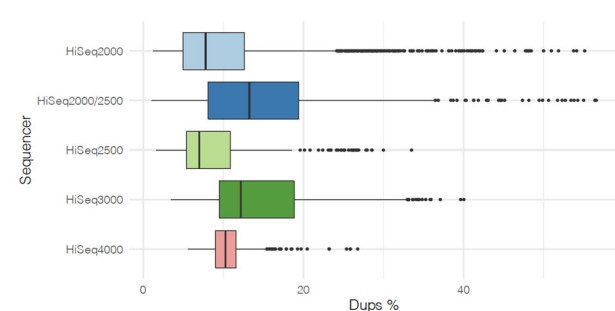

**i) Percentage of mapped reads**

**ii) Percentage of duplicated reads**

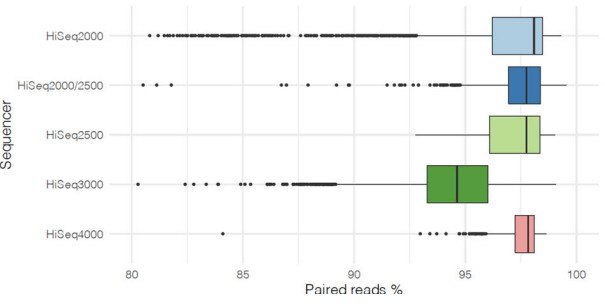

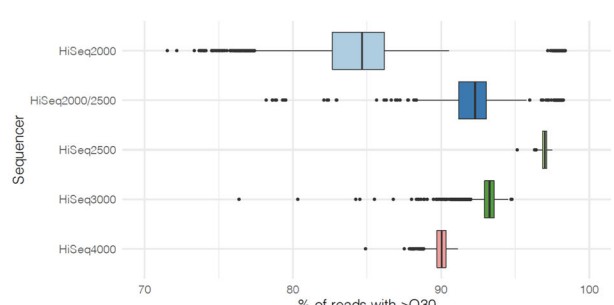

**iii) Percentage of paired reads**

**iv) Quality of reads (based on Q30 score)**

**Fig. 4 | Comparison of WES CRAM quality metrics.** We compared the CRAM quality metrics across **a** sequencing centers (Seq_center); and **b** sequencing platforms (Sequencer). *N* for each of these 8 plots (**a** i to iv), **b** (i to iv)) all equals 20,504 subjects. Quality metrics included (i) Percentage of mapped reads, (ii) Percentage of duplicated reads, (iii) Percentage of paired reads, and iv) Quality of reads based on Q30 score. For each box plot, the center line represents the median value, the minimum of the whisker represents the 1st quantile (25th percentile), and the maximum of the whisker represents the 3rd quantile (75th percentile). Source data are provided as a Source Data file.

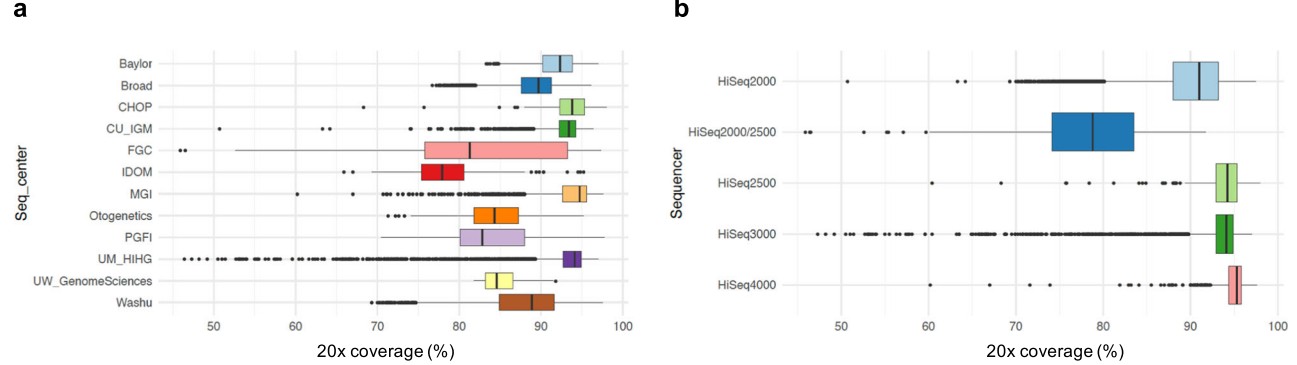

**Fig. 5 | Comparison of ×20 coverage of all the WES BAMs/CRAMs.** We compared the ×20 coverage (defined by the percentage of bps with reads or more within the sample-specific capture regions) first by **a** Sequencing centers, then by **b** Sequencers. *X* axis show the "×20 coverage" in percentages. *N* for **a** and **b** are both

20,504 subjects. For each box plot, the center line represents the median value, the minimum of the whisker represents the 1st quantile (25th percentile), and the maximum of the whisker represents the 3rd quantile (75th percentile). Source data are provided as a Source Data file.

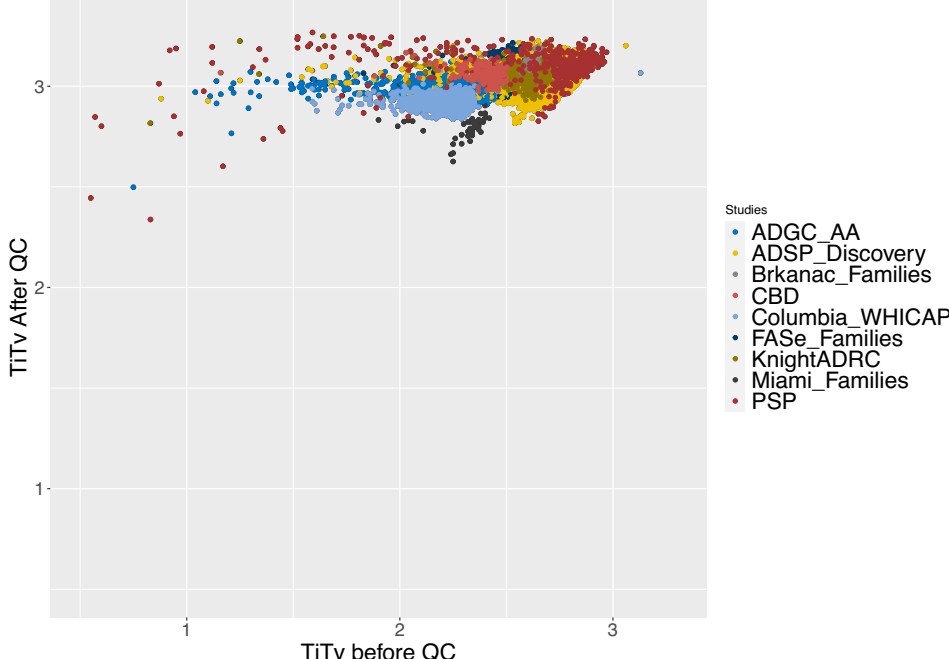

**Fig. 6 | Comparison of the Ti/Tv ratio of exonic variants before and after QC.** Ti/Tv ratio is the ratio of transition (Ti) to transversion (Tv) SNPs. *X* axis shows the Ti/Tv ratio before QC, while the *Y* axis shows the Ti/Tv ratio after QC. The average Ti/

Tv ratio increases from 2.53 to 3.03 after the QC process. This post-QC Ti/Tv ratio is similar to reports in previous findings. Source data are provided as a Source Data file.

Instructions page (https://dss.niagads.org/documentation/applying-for-data/application-instructions/) on how to submit a Data Access Request and access data.

## Discussion

In this study, we developed a new bioinformatics approach, VCPA-WES, to joint-call WES samples sequenced using multiple capture kits from different sequencing providers. The procedure has been successfully applied to a total of 20,504 exomes gathered through the collaborative network of the ADSP, resulting in the generation of the world's largest publicly available AD WES data collection and joint-call pVCF to date. Annotated and QC-ed following ADSP protocols, this high-quality WES pVCF with ~7.5 million SNVs and >700,000 indels is publicly available at NIAGADS DSS for qualified investigators worldwide.

Our approach was significantly different from what was used in Holstege et al.[19]. Different software was chosen for marking duplicated reads, and the strategies to call variants at the capture regions were

different; even if the same software was used, parameters and versioning were different (Supplementary Table 13). The sequence or joint-genotype data reported by this group was not publicly available, making it challenging to compare pipeline effects on variants called in both datasets.

Joint-calling WES samples based on different capture kits poses several challenges due to the nature of the kits. Although each capture kit was designed primarily based on exonic regions (> 91% of the capture regions per capture kit were covered by Ensembl exons, (Table 3, Section Characteristics of Capture kits), they were also designed based on different genome builds and gene annotations, therefore resulting in substantial differences in the captured contents (average Jaccard similarity score of ~0.6 per capture kit vs all other kits, (Fig. 3, Section Capture kits comparison). In order to successfully harmonize/joint-call all of the data without any systematic bias, a uniform bioinformatics pipeline, together with the standardization of capture kit target region definitions, is critical.

In our calling strategy, we first lifted over all capture kits using the same protocol to GRCh38 if they were not of this genome build. We then processed all WES samples using a single analysis pipeline: VCPA-WES (Section VCPA for WES processing). We did not use capture region definitions to limit variant calls when we generated gVCF or joint-called pVCFs. Capture region definitions were only used in the QC steps to identify high-quality variant and genotype calls. There are two advantages of this approach. First, in the future, when additional samples on different capture kits need to be incorporated, we can reuse the gVCFs of these old samples without reprocessing gVCFs. Second, we can retain variants and genotype calls that are either outside the target regions but still have good quality or in regions that are not targeted by all capture kits. Indeed, if we only look at regions that are targeted by all capture kits, we are left with only about 45% of the unions.

The genotype data generated by this tailor-made bioinformatics strategy are high-quality. We have shown in Figs. 4 and 5 that there is no systematic bias attributed to the sequencing centers and sequencers on the CRAM quality (Section Data quality—WES CRAMs) (except for Q30 scores and 20× coverage). This shows that even though the WES data were sequenced in different ways, these experimental artifacts could be greatly reduced with a carefully designed data processing pipeline.

Next, we evaluated the quality of the variants using the GATK VQSR score, Ti/Tv ratio (Fig. 6), and via our in-house GCAD/ADSP QC protocol (Section Data quality—variants). Overall, ~97% of variants were labeled PASS by GATK. After variant-level QC was performed, the Ti/Tv ratio in exonic regions in our studies was >3. This is similar to what was previously reported, indicating that the data doesn't contain many false positives caused by random sequencing errors. In terms of study-capture-specific variants, using a multi-step QC protocol as shown in Table 4 (exclude variants with high missingness rate, excessive heterozygosity, high read depth, etc.), >91% of variants have successfully met our QC criteria, are of good quality, and can be used in subsequent analyses.

Lastly, we performed genotype concordance analyses on the set of overlapping samples found in both the previously published ADSP-Discovery WES data set and this newly joint-called WES data set. Around 11,000 samples were used for the analyses (Section Genotype Concordance between two different callers on a set of overlapping samples). Even though the two datasets were called using different callers (ATLAS2 vs GATK), >1.4 million variants were called in both datasets, comprising 15 billion genotypes. Overall concordance was 99.43%. Variants that were not concordant may be located in genomic regions that are difficult to be sequenced. This shows that, despite the complexity involved in creating this new, much larger, joint-called WES data set, the innovative bioinformatics strategy allowed us to produce a data set with high-quality genotypes.

The 8.16 million variants in this WES data set span across 28,579 transcripts (Section Annotation results). We annotated every variant based on the most damaging VEP predicted consequence (Supplementary Fig. 5a). The top ten most damaging consequence categories included missense variants, upstream/downstream gene variants, synonymous variants, and 3'UTR variants. Meanwhile, 15.5% of the variants have a normalized CADD score >20, meaning that these variants are among the top 1% of deleterious variants in the human genome (Supplementary Fig. 5b). These results showcase that the ADSP annotation pipeline we developed is very well capable of annotating both WGS and WES data.

In conclusion, the VCPA-WES bioinformatics pipeline, publicly available at https://bitbucket.org/NIAGADS/vcpa-pipeline/src/master/, together with the QC and annotation protocol GCAD developed, enable us to generate a high-quality AD-specific WES data set containing over 8 million variants on 20,504 samples. The pipeline works well when joint-calling any WES data (of different phenotypes) that are sequenced in different batches using different sequencing machines and capture kits.

This harmonization approach to alignment and variant calling has minimized the impact of different experimental and analytical pipelines on this valuable dataset, which is, by far, the largest publicly available AD WES data set. It is available at NIAGADS DSS: https://dss.niagads.org/datasets/ng00067/. Qualified and approved investigators can apply to access and download the data for various research purposes.

# Methods
## Data description
**Sample selection.** The data set consists of 20,504 samples across nine studies (Table 1). Approximately half of the samples are from the ADSP-Discovery (part of the ADSP case/control data). Case-control statuses were defined using the NINCDS-ADRDA (National Institute of Neurological and Communicative Disorders and Stroke, and the Alzheimer's Disease and Related Disorders Association) criteria[20] or NIA-AA (National Institute on Aging Alzheimer's Association) criteria[21,22]. GCAD reached out to ADGC/ADSP PIs in the collaborative network and received 9847 additional WES samples from eight different studies. None of the studies prioritize one sex over the other when doing recruitment or study design. Sex and/or gender of participants were determined based on self-report. Table 1 contains counts of samples that have passed through QC (see the section on Sample-level quality assurance checks for details).

## Genome sequencing and capture kit information
Libraries were constructed from sample DNA with PCR amplification. Sequencing was performed across fourteen sites using different combinations of Illumina sequencing platforms and capture kits. Several studies (e.g., ADSP-Discovery) used multiple kits. While other studies used a single capture kit for all samples in their study design (e.g., Roche Nimblegen's VCRome v2 kit was used in ADSP-Discovery, PSP, and Knight ADRC studies). The details are summarized in Table 1.

## Demographics
Phenotype information, such as disease status (AD and other dementia case or cognitively intact control), self-reported race/ethnicity, sex, and age-at-onset (cases)/age-at-last exam (controls), as well as the number of APOE ε2/ε3/ε4 alleles per individual, were obtained from the phenotype data shared by the data contributors. The demographics of this dataset are summarized in Table 2. Altogether, there are a total of 9955 cases (mean ± SD: 75.8 ± 8.9 years old) and 9717 controls (mean ± SD: 81.5 ± 8.1 years old). 60% of the cases were female, with a similar sex-gender ratio for controls. 44% of the cases (and 23% of the controls) have ≥1 APOE ε4 alleles, which is a known genetic risk factor for AD.

## Methods
**Integrating multiple WES capture kits.** As summarized in Table 1, 10 different capture kits were used for sequencing samples across nine studies. These capture kits were manufactured over the years by three different vendors (Illumina, Agilent, and Roche) and had substantial differences in kit contents, as the capture kits were generated based on different genomic annotation databases (e.g., Ensembl[8] on different reference genome builds [GRCh36, GRCh37]). Whenever possible, capture kit annotation files (in BED format) were received directly from the data contributors. If not, GCAD downloaded the original capture kit annotation files from the vendor's website. Note that these files contain genomic coordinate information and do not include the exact sequences of the designed regions. If GRCh38 information of capture kits is not available, we performed UCSC liftOver[23] converting coordinates to GRCh38. All regions were further

**Table 4 | Variant quality per study-capture combination**

| Study | Capture Kit info | % of variants that are monomorphic | % of non-monomorphic sites, in capture kits | Within the capture kits, % with GATK PASS | Within the capture kits, % with VFLAG 0 |
|---|---|---|---|---|---|
| MIAMI family | Agilent WES v3 capture region | 97.65 | 63.22 | 97.56 | 91.70 |
| MIAMI family | Agilent WES v4 capture region | 97.88 | 57.95 | 95.29 | 72.50 |
| CBD | Agilent WES v5 capture region | 94.20 | 68.74 | 98.09 | 95.30 |
| FASe family | Agilent WES v5 capture region | 93.60 | 70.88 | 97.57 | 91.84 |
| ADGC AA | Agilent WES v6 capture region | 71.92 | 73.71 | 97.56 | 94.81 |
| Knight ADRC | IDX xGen Exome Whole-Exome Research Panel v1.0 w/Custom Spike-in Baits | 95.90 | 47.79 | 97.44 | 95.48 |
| ADSP-Discovery | Illumina Rapid Capture Exome (ICE) kit | 77.87 | 54.20 | 97.90 | 93.87 |
| FASe family | Nimblegen VCRome sequencing w/Custom Spike-in Baits | 96.06 | 46.57 | 96.73 | 93.86 |
| Knight ADRC | Nimblegen VCRome sequencing w/Custom Spike-in Baits | 91.67 | 47.97 | 97.56 | 95.31 |
| ADSP-Discovery | Roche Nimblegen's VCRome v2.1 | 69.64 | 66.22 | 98.51 | 94.69 |
| FASe family | Roche Nimblegen's VCRome v2.1 | 95.91 | 62.59 | 96.95 | 91.74 |
| PSP | Roche Nimblegen's VCRome v2.1 | 92.47 | 59.10 | 95.60 | 84.13 |
| Brkanac | Roche SeqCap EZ Exome Probes v3.0 Target Enrichment Probes | 97.17 | 64.59 | 97.45 | 91.17 |
| Columbia WHICAP | Roche SeqCap EZ Exome Probes v3.0 Target Enrichment Probes | 67.19 | 78.37 | 97.39 | 93.90 |

The QC protocol enables us to look for variants found across studies as well as those that are study-specific. On average, 97.26% of variants have a GATK PASS across study-capture combinations. 91.45% of variants within the designed capture kit per each study-capture combination are labeled as good quality.

combined as a single BED file that contains the union of the capture kits' genomic intervals (with flanking ± 7 basepairs [bps]). The BED files for individual capture kits, as well as the BED file containing all the captures' intervals, are available in NIAGADS DSS (https://dss.niagads.org/wp-content/uploads/2021/08/gcad.wes_.20650.VCPA1_.1.2019.11.01.targetregions.zip?x78736).

**Processing WES using VCPA at the individual sample level.** VCPA, a BWA/GATK-based pipeline[24,25], was developed by GCAD and the ADSP and optimized for processing large-scale, short-read WGS data[5]. To adopt VCPA for ADSP WES data processing, GCAD followed GATK Best Practices[26] with the following steps modified to accommodate for the use of multiple WES capture kits (VCPA-WES, Fig. 1). Instead of calling variants limited to the capture regions per sample[3,4], VCPA-WES keeps all detected variants (same as that of VCPA-WGS, but different than other WES based pipelines[3,4]), as we envision that (1) the research community will use the joint-called VCF for different kinds of analyses (i.e., one project will select a few studies for its analyses, but another project might pick different studies); and (2) Any WES datasets in the future may use different capture kits. Compared to VCPA-WGS, VCPA-WES (at the individual sample level) differs in the following steps (components highlighted with a "star" in Fig. 1):

- Coverage calculation−the 20× coverage metric (i.e., percentage of bps with 20 reads or more)[4,9–13] was calculated on regions that were included in the capture kits only. No bad-quality reads were filtered prior to assessment.
- Variant evaluation−Ti/Tv ratio (ratio of the number of transitions to the number of transversions) and counts of SNPs/indels were calculated on regions that were included in the capture kits only.

**Joint-genotype calling of 20,504 WES samples.** All WES samples were jointly called using GATK4.1.1 to create a joint genotype called project-level VCF (pVCF). This included these major steps:

- VCPA-WES performs joint-genotype calling across samples on all possible variants called (not limited to captures). CombineGVCF and GenotypeGVCF are the same in both VCPA-WGS and VCPA-WES pipelines. gVCFs of all 20,504 samples were combined in parallel across 5,000 genomic windows/regions across all the chromosomes.
- Generating the VQSR model−a Variant Quality Score Recalibration (VQSR) indicator is used for defining qualities of variants via a machine-learning model. Only variants that were called within any of the capture kits were used to build the VQSR model. VQSR model classify the variants in any capture kits with different quality scores (VQSR tranches). This step is modified upon VCPA-WGS (Fig. 1).
- Applying the VQSR model−the trained VCPA-WES VQSR model (on variants within any capture kits) was applied to all the autosomal chromosomes, as well as chromosomes X and Y, and mitochondria.

**Sample-level quality assurance (QA) checks.** Three quality assurance checks were applied prior to joint-genotype calling to identify problematic samples:

- *SNV concordance check* with existing SNP array genotypes to identify possible sample errors. Using verifyBamID[27] to compare between SNP array data and WES BAM files, samples with concordance <0.95 were excluded.
- *check* for variants outside PAR region to identify possible sample swaps or misreporting. Using PLINK, samples with *F* statistics < 0.2 or > 0.8.
- *Contamination check* for possible sample swaps. Using verifyBamID[27] to calculate the concordance estimate between the array genotypes and the GRCh38-mapped BAM file. A sample is potentially contaminated if the CHIPMIX value is <0.05.

In total, we dropped 55 samples that failed the SNP concordance check, 41 samples that were recorded with the incorrect sex, and 211 samples that failed the contamination check. An additional 266 samples were dropped due to consent issues, resulting in a call set containing 20,504 samples.

**Quality control (QC) protocol for WES samples.** The GCAD quality control (QC) pipeline uses a modified protocol originally developed by the ADSP QC Working Group on WGS[14] and includes several major components: (1) pre-QC quality checks; (2) variant-level QC; (3) sample-level QC; and (4) post-QC quality checks. These steps are applied to both SNVs and indels. We implemented variant-level QC to SNVs and indels in the project-level VCFs. Data were stratified into sequencing subsets based on the capture kit, sequencing assay, and sequencing center. We applied filters in the following order within sequencing subsets, resulting in the exclusion of (1) variants outside of designated capture regions specific to the capture kit used on a sample; (2) variants failing GATK quality assessment (those without "PASS" or in a VQSR Tranche of 99.5% or more extreme); monomorphic variants; (3) variants with a high missingness rate (≥20%); (4) variants with excessive heterozygosity and (5) variants with high average read depth (>500x). We estimated allelic read ratios (ABHet) among heterozygotes for each variant as an optional metric for excluding variants with extreme deviations from the expected ABHet of 0.5.

Additionally, we generated several metrics within ancestry groups. Phenotype data on all participants included self-reported race and ethnicity values corresponded almost exactly with membership in specific data sources/originating studies. To identify preliminary ancestry groupings (NHW/AFA/CHI), we partitioned the dataset by data source/originating study and grouped them by the originating study's recruitment criteria based on self-reported race/ethnicity (e.g., studies recruiting self-identifying African American subjects with AFA, studies recruiting in Caribbean Hispanic communities with CHI). Source datasets containing more than one self-identifying race/ancestry group with at least 200 participants were partitioned, and the subsets added to each preliminary ancestry grouping. Separately for NHW, AFA, and CHI groupings, we combined the samples in each with 1000 Genomes (1kG) reference populations and sampled ~20,000 variants in low linkage disequilibrium (LD) within the data ($r^2 < 0.2$) that overlapped the ancestry grouping dataset and the 1kG reference (MAF > 0.01). We then estimated population substructure within ancestry grouping through principal components analysis on these ~20,000 variants using EIGENSOFT[28,29]. This was used to broadly confirm sample set clustering within expected 1kG reference populations and to filter out outliers within each ancestry grouping. Participant samples were defined as NHW if they clustered in principal components (PCs) 1 and 2 within 3 SD of the European (EUR) populations in 1kG, which led to the exclusion of 32 outlier samples; as AFA if they clustered within 3 SD of EUR and African (AFR) populations in 1kG or between EUR and AFR clusters, which led to the exclusion of 29 samples; and as CHI if they clustered within 3 SD of EUR and AFR populations or between those clusters and subsets of Puerto Rican, Peruvian, and Mexican, and Amerindian ancestry individuals in 1kG (no samples were excluded in this subset). Excluded samples were not reincluded in other ancestry groupings because of differences by sequencing center/assay and capture kit.

Among the metrics estimated within each ancestry group were departure from Hardy-Weinberg Equilibrium (HWE) among controls or excess heterozygosity for datasets with related samples. Either of these metrics may be used as potential exclusion criteria by end-users of the data.

We also explored sample-level QC criteria and evaluated multiple filters to further exclude potential low-quality samples. We estimated

multiple quality metrics within each sample including (1) counts of singleton/doubleton variant calls (to identify an excess of private variants); (2) genotype missingness rate within the sample; (3) Transition/Transversion (Ti/Tv) ratio (for SNVs only); (4) heterozygosity-to-homozygosity ratio across all variants within individuals; and (5) the mean within-sample read depth. Samples were considered for exclusion if their values for any of these criteria were greater than 6 SD from the mean value.

**Genotype concordance analyses with the previously published ADSP-Discovery WES data.** Genotype calls generated by VCPA in this data set (20,504 samples) include samples that were part of the ADSP-Discovery data set[1]. To provide a comparison of genotype quality, we examined the concordance between genotype calls using the ATLAS2 approach[30] on 10,786 samples that overlapped between the two sets. The previous genotype calling was conducted based on GRCh37, which was lifted to GRCh38 using liftOver[23]. For these analyses, genotype concordance was defined as identical genotype calls (including missing genotypes) between the two call sets. Concordance was calculated by sample, by variant, and overall. Because VCPA employs a joint-calling approach, we also investigated the impact of the capture kit coverage on genotype concordance under the hypothesis that limited coverage in additional samples of the current data set could alter the quality control metrics.

**Annotation protocol for WES samples.** Variants were annotated using our published annotation pipeline with updated resources (VEP 98[16], CADDv1.4[17], SnpEffv4.1k[31]) in GRCh38, described elsewhere[15]. Briefly, we assign a "most damaging consequence" via a custom prioritization routine that down-weights non-coding transcripts or transcripts flagged as undergoing nonsense-mediated decay.

To compare the distribution of coding variants against publicly available WES annotation resource, we downloaded gnomADv2.1 GRCh38/hg38 lifted-over variant dataset from: https://gnomad.broadinstitute.org/downloads#v2-liftover .

**Recommended post-QC processing for analysis.** To prepare data for analysis, we recommend a number of filtering steps. Initially, we recommend the removal of samples based on an available list of unintentional duplicates across sequencing studies and subsets, as well as a list of intentional replicates included to perform comparisons between sequencing experiments. We provide recommendations to end-users as to which samples should be kept or excluded, prioritizing (a) completeness of genotype data, (b) data collection with which the replicate or duplicate was originally ascertained, and (c) sample size of the dataset to which the replicate or duplicate belongs, prioritizing keeping samples in smaller subsets or studies. We then recommend identifying and removing samples flagged for not having a GATK FILTER value of "PASS"; having all genotypes set to missing after genotype-level QC; being monomorphic within subset; having a low call rate (<0.8) across all subsets; having extremely high read depth (DP > 500) indicating potential read misalignment; and having allelic read ratios >0.75 or <0.25. Due to subset sizes, we do not recommend filtering on Hardy-Weinberg disequilibrium or excess heterozygosity. Finally, for association analyses including data across subsets, we recommend using one of multiple approaches, including (a) filtering on variants not failing in all subsets and performing joint analysis across subsets adjusting for population substructure estimated across all samples, and indicator variables for study subset (this can be modified to perform analysis within ancestry groups across subsets); or (b) filtering failing variants out of each subset, estimating population substructure within subset, performing association analysis within subset, and then performing meta-analysis across subsets to combine results (this approach can include random-effects meta-analysis across ancestry groups to get cross-ancestry associations). While this approach did not directly address every potential sequence quality issue individually, this filtering strategy minimized most notable quality issues (including differential retention of low read depth genotype calls by genotype) by excluding off-target variants.

### Reporting summary
Further information on research design is available in the Nature Portfolio Reporting Summary linked to this article.

## Data availability
All CRAMs, gVCFs generated by GATK4.1.1, and QC-ed pVCFs of the abovementioned ADSP WES data set are available in the NIAGADS Data Sharing Service (DSS) (NG00067.v3), together with pedigree structures for family studies and phenotypes that were harmonized according to ADSP protocols. The WES target regions (from GRCh36, GRCh37) now lifted to GRCh38 for analyses are available as well. Qualified investigators can access these data with a submission request and approval from the NIAGADS Data Access Committee managed by independent NIH program officers. Data can be downloaded through the DSS portal. More information about the data set can be found on the data set page, NG00067. See the Application Instructions page (https://dss.niagads.org/documentation/applying-for-data/application-instructions/) on how to submit a Data Access Request and access data. Source data are provided in this paper.

## Code availability
VCPA-WES code is publicly accessible at https://bitbucket.org/NIAGADS/vcpa-pipeline/src/master/.

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

## Acknowledgements

Data for this study were prepared, archived, and distributed by the National Institute on Aging Alzheimer's Disease Data Storage Site (NIAGADS) at the University of Pennsylvania (U24-AG041689), funded by the National Institute on Aging. For complete acknowledgments for each cohort used in this publication, please refer to Supplementary Notes for details.

## Author contributions

Y.Y.L., A.C.N., Y.-F.C. and W.S.B. conceived and designed the experiments. Y.Y.L., Y.-F.C., N.W., K. H.-N., M.S. and W.S.B. performed data analyses. Y.-F.C., O.V., P.G., L.Q. carried out the data production under the supervision of Y.Y.L., A.C.N. and E.M. A.B.K., L.C. and H.N. manages phenotypes for these samples. Y.Y.L., H.L., K.C., W.-P. L. and L.-S.W. wrote and edited the manuscript. G.D.S., L.-S. W., E. M., A.C.N., A.D., R.P.M., M.P.-V., C.C., Z. B., S.S., L.F. and J.H. secured funding. All authors read and approved the manuscript.

## Competing interests

The authors declare no competing interests.

## Additional information

[1]Penn Neurodegeneration Genomics Center, Department of Pathology and Laboratory Medicine, Perelman School of Medicine, University of Pennsylvania, Philadelphia, PA, USA. [2]Department of Biostatistics, Epidemiology, and Informatics, Perelman School of Medicine, University of Pennsylvania, Philadelphia, PA, USA. [3]Dr. John T. Macdonald Foundation Department of Human Genetics, Miller School of Medicine, University of Miami, Miami, FL, USA. [4]The John P. Hussman Institute for Human Genomics, University of Miami, Miami, FL, USA. [5]Department of Population and Quantitative Health Sciences, Case Western Reserve University, Cleveland, OH, USA. [6]Department of Genetics and Genome Sciences, School of Medicine, Case Western Reserve University, Cleveland, OH, USA. [7]Department of Medicine, UMass Chan Medical School, Boston, MA, USA. [8]Department of Medicine (Biomedical Genetics), Boston University Chobanian & Avedisian School of Medicine, Boston, MA, USA. [9]Department of Biostatistics, Boston University School of Public Health, Boston, MA, USA. [10]Boston University School of Medicine, Boston, MA, USA. [11]The Glenn Biggs Institute for Alzheimer's and Neurodegenerative Diseases, University of Texas Health Sciences Center, San Antonio, TX, USA. [12]Department of Psychiatry and Behavioral Sciences, University of Washington, Seattle, WA, USA. [13]Washington University School of Medicine, St. Louis, MO, USA. [14]Department of Neurology, Taub Institute for Research on Alzheimer's Disease and the Aging Brain and the Gertrude H. Sergievsky Center, Columbia University and the New York Presbyterian Hospital, New York, NY, USA. [15]Department of Neurology, Boston University School of Medicine, Boston, MA, USA.
✉e-mail: yyee@pennmedicine.upenn.edu; lswang@pennmedicine.upenn.edu

## Alzheimer's Disease Sequencing Project

Sven van der Lee[16], Adam English[17], Divya Kalra[17], Donna Muzny[17], Evette Skinner[17], Harsha Doddapeneni[17], Huyen Dinh[17], Jianhong Hu[17], Jireh Santibanez[17], Joy Jayaseelan[17], Kim Worley[17], Richard A. Gibbs[17], Sandra Lee[17], Shannon Dugan-Perez[17], Viktoriya Korchina[17], Waleed Nasser[17], Xiuping Liu[17], Yi Han[17], Yiming Zhu[17], Yue Liu[17], Ziad Khan[17], Congcong Zhu[10], Fangui Jenny Sun[10], Gyungah R. Jun[10], Jaeyoon Chung[10], John Farrell[10], Xiaoling Zhang[10], Eric Banks[18], Namrata Gupta[18], Stacey Gabriel[18], Mariusz Butkiewicz[5,6], Penelope Benchek[5,6], Sandra Smieszek[5,6], Yeunjoo Song[5,6], Badri Vardarajan[14], Christiane Reitz[14], Dolly Reyes-Dumeyer[14], Giuseppe Tosto[14], Phillip L. De Jager[14], Sandra Barral[14], Yiyi Ma[14], Alexa Beiser[9], Ching Ti Liu[9], Josee Dupuis[9], Kathy Lunetta[9], L. Adrienne Cupples[9,36], Seung Hoan Choi[9], Yuning Chen[9], Jesse Mez[15], Ashley Vanderspek[19], M. Arfan Ikram[19], Shahzad Ahmad[19], Kelley Faber[20], Tatiana Foroud[20], Elisabeth Mlynarski[21], Helena Schmidt[22], Reinhold Schmidt[22], Brian Kunkle[3,4], Farid Rajabli[3,4], Gary Beecham[3,4], Jeffrey M. Vance[3,4], Larry D. Adams[3,4], Michael Cuccaro[3,4], Pedro Mena[3,4], Briana M. Booth[23], Alan Renton[24], Alison Goate[24], Edoardo Marcora[24], Adam Stine[25], Michael Feolo[25], Lenore J. Launer[26], Daniel C. Koboldt[27], Richard K. Wilson[27], Cornelia van Duijn[28], Najaf Amin[28], Manav Kapoor[29], William Salerno[29], David A. Bennett[30], Li Charlie Xia[31], John Malamon[32], Thomas H. Mosley[33], Claudia Satizabal[11], Jan Bressler[11], Xueqiu Jian[11], Alejandro Q. Nato Jr[34], Andrea R. Horimoto[34], Bowen Wang[34], Bruce Psaty[34], Daniela Witten[34], Debby Tsuang[34], Elizabeth Blue[34], Ellen Wijsman[34], Harkirat Sohi[34], Hiep Nguyen[34], Joshua C. Bis[34], Kenneth Rice[34], Lisa Brown[34], Michael Dorschner[34], Mohamad Saad[34], Pat Navas[34], Rafael Nafikov[34], Timothy Thornton[34], Tyler Day[34], Jacob Haut[2], Jin Sha[2], Nancy Zhang[2], Taha Iqbal[2], Laura Cantwell[1], Yi Zhao[1], Jennifer E. Below[35], David E. Larson[13], Elizabeth Appelbaum[13], Jason Waligorski[13], Lucinda Antonacci-Fulton[13] & Robert S. Fulton[13]

[16]Amsterdam UMC, Amsterdam, The Netherlands. [17]Baylor College of Medicine, Houston, TX, USA. [18]Broad Institute of MIT and Harvard, Cambridge, MA, USA. [19]Erasmus Medical University, Rotterdam, Netherlands. [20]Indiana University, Fort Wayne, IN, USA. [21]Johnson & Johnson, Horsham, PA, USA. [22]Medical University Graz, Graz, Austria. [23]MITRE, McLean, VA, USA. [24]Mount Sinai School of Medicine New York, New York, NY, USA. [25]National Center Biotechnology Information, Bethesda, MD, USA. [26]National Institute on Aging, Bethesda, MD, USA. [27]Nationwide Children's, Columbus, OH, USA. [28]Oxford University, Oxford, UK. [29]Regeneron, Tarrytown, NY, USA. [30]Rush University, Chicago, IL, USA. [31]Stanford University, Stanford, CA, USA. [32]University of Colorado, Boulder, CO, USA. [33]University of Mississippi, Oxford, MS, USA. [34]University of Washington, Seattle, WA, USA. [35]Vanderbilt University, Nashville, TN, USA. [36]Deceased: L. Adrienne Cupples.

