## [Peer Review File · Nature Communications]

REVIEWER COMMENTS

Reviewer #1 (Remarks to the Author):

This paper reports a pipeline for processing of WES data generated on a range of platforms, with different capture kits in different research centers, addressing a well-known issue in the field. The pipeline has been applied to a large collection of WES datasets on AD study cohorts. The paper describes the pipeline, and descriptively compares data between platforms, sequencing centers and capture kits. The focus of the manuscript is on describing the datasets obtained through this pipeline, as the title correctly reflects. No findings of relevance to the phenotype of interest are presented.

1. While the paper refers to the pipeline as a new computational framework, the data have been made available to the scientific community in February 2020 (line 93-94). If the pipeline is in fact a recent new addition that has substantially improved the data at this portal, this should be more clearly stated. If this pipeline was already applied to the data that were shared publicly in February 2020, then that limits the novelty of the current manuscript.

2. How does the new bioinformatics pipeline compare to other more recent approaches in the AD field or beyond? E.g. recently, a much larger effort to jointly analyze WES/WGS datasets derived from multiple centers/platforms/capture kits has been performed in the AD field (Holstege et al Nature Genetics 2022). This study includes, amongst others, the data presented in the current manuscript. How does the computational framework compare to the methods in Holstege et al? In that paper, specific attention was drawn to technical batch effects, e.g. due to use of restriction enzymes in certain capture kits, which may be affected by mutations in restriction sites. How are this and similar technical causes of batch effects addressed in the pipeline presented here?

3. The methods section only reports the bioinformatics pipeline that was followed to process the WES data. It would be helpful to also include a description on how the results were analyzed, e.g. how was the Jaccard similarity measure computed?

4. The authors conclude that differences in sequencing platform or center did not affect processed CRAMs or 20x coverage. Was this conclusion based on eyeballing, or was this formally tested? The latter would be preferable, in which case the statistics used should also be included in the methods section.

Reviewer #2 (Remarks to the Author):

This paper describes a joint calling strategy for the meta-analysis of Alzheimer's Disease exomes, considering using capture kits and detecting batch effects during the QC process. The data has been deposited. Although the generation of genomic resources for the community is appreciated, joint calling and sample QC is now standardized through existing tools. This is especially true for a GATK-based pipeline and is usually presented as part of an analysis paper rather than a standalone paper. Examples of such datasets from the last five years include the gnomAD project, the Autism Sequencing Consortium, Deciphering Developmental Disorders, TOPMED, and UK10K studies, all of which handle multiple capture kits or include whole-genome and whole-exome sequencing data. Therefore, while this paper can still be valuable (as shared community resources are), a few areas need additional clarity and detail to maximize value for the scientific community.

Tables and Figures:

Table 1: Can you provide the number of cases and controls per study/capture to understand better how the study is distributed across captures?

New figures: Because the study describes a modified calling pipeline, can a figure be added showing the step-by-step workflow of processing the data? Because the pipeline largely uses the existing "VCPA" pipeline, it would help disentangle the genome and exome-specific components. I don't think the details of the VCPA pipeline should be presumed.

Demographic information, especially related to genetic ancestry, should be reworded slightly in consideration of <https://nap.nationalacademies.org/resource/26902/interactive/>.

More importantly, it would be helpful to have a PCA plot highlighting diversity within this sample. Separately, it would be helpful to see a plot showing the relatedness within this sample as an additional data set descriptor.

Analysis:

From Figure 1, it appears that the overall overlap between captures is very low (0.586) when they should all target the exome. Can you provide more context as to why that is and how that affects the overall use of the data? Furthermore, what does this mean for capturing the coding region? Rather than focusing on the capture kit, we can look at the coverage of the coding region instead.

Is there a reason why the 20x coverage is used to determine the quality of these data?

Can you compare the distribution of coding variants (syn, mis, PTVs) in these data compared to other published data (like gnomAD)?

As this document is a "Data Descriptor," the overall text is light on analyses. Given the observation of batch effects (by capture), can you highlight how you would recommend that this data be analyzed for gene discovery? A useful "QC" metric compares the rates of synonymous variants between cases and controls on a gene-by-gene basis. This ensures that the QC is performed adequately. Can you provide this figure as an additional bit of analysis?

In that vein, the absolute number of coding variants (nhets, nhoms, nhomvar) in each cohort are useful metrics to see how comparable the sample is before and after QC. Can you provide that as a separate figure in the text (split by cohort, capture)?

Code:

Usually, I do not over-emphasize code in a review of a paper, but because this is a resource paper, I took a deeper look. The code and the Gitbucket repo should be directly highlighted in the text (<https://bitbucket.org/NIAGADS/vcpa-pipeline/src/master/VCPA/>). From a look at the repo, most scripts are from an earlier iteration focusing on whole-genome sequence data (2018?). It will be helpful to highlight the workflow and scripts that are specific to the exome data and how to use them. The Supplementary Note includes two commands from GATK, and I would prefer that to be directly in the codebase. Furthermore, I cannot access the wiki, which might be helpful in understanding how to run the code.

Minor:

There needs to be some editing and clarification of the acronyms in the text. A few examples include QC'ed and VCPA in the background.

Can a variant-only file with allele frequencies like gnomAD be provided to the community?

Reviewer #3 (Remarks to the Author):

The Data Descriptor manuscript by Leung et al. describes an expanded cohort of whole exome sequencing data from over 20,000 individuals with a diagnosis of Alzheimers disease.

The manuscript is comprehensive, well written and describes an exceptional resource that is important for the research community.

I have a series of minor questions and comments about the manuscript that are focused on the improvement of the manuscript.

1. It would be informative to add the number of samples in the study that used each capture technologies to table 3.
2. For the comparison of 20x sequence coverage (Figure 3 and text), it is unclear filtering of reads if any was performed prior to assessment (ie were duplicate reads, unpaired reads and/or >Q30 mapping score reads excluded).
3. One area which is unclear in the current manuscript is the quality of genotype calls across sites. I think the manuscript would benefit from an assessment per variant site of the genotype call rate when genotype calls with GQ<20 (or some justifiable threshold) are set to missing - a histogram across variant sites would be informative (potentially also stratified by capture platform). As this will really get to the crux of the challenge with this call set, which is that due to the differences in capture technologies one would not expect high quality data for all samples at all sites.
4. The statement in the discussion that the dataset is 'free of batch effects' is an overstatement and I am sure that it would be possible to identify batch effect by call-rate (see above comment), the authors may want to revise this statement to reflect that they have employed a harmonisation approach to alignment and variant calling to minimise the impact of different analytical pipelines.

REVIEWER COMMENTS

Reviewer #1 (Remarks to the Author):

This paper reports a pipeline for processing of WES data generated on a range of platforms, with different capture kits in different research centers, addressing a well-known issue in the field. The pipeline has been applied to a large collection of WES datasets on AD study cohorts. The paper describes the pipeline, and descriptively compares data between platforms, sequencing centers and capture kits. The focus of the manuscript is on describing the datasets obtained through this pipeline, as the title correctly reflects. No findings of relevance to the phenotype of interest are presented.

1. While the paper refers to the pipeline as a new computational framework, the data have been made available to the scientific community in February 2020 (line 93-94). If the pipeline is in fact a recent new addition that has substantially improved the data at this portal, this should be more clearly stated. If this pipeline was already applied to the data that were shared publicly in February 2020, than that limits the novelty of the current manuscript.

Response: We thank the reviewer for the comment. We have clarified in the manuscript that there are two novel contributions to this paper: 1) the VCPA-WES pipeline that was designed to handle a variety of capture kits used in sequencing, and 2) the description of the largest publicly available WES data for AD, both of which have not been presented or described in any publication before. The data has only been available to the scientific community in 2020 via a data sharing platform (NIAGADS DSS), but has not be presented in any manuscript/publication.

2. How does the new bioinformatics pipeline compare to other more recent approaches in the AD field or beyond? E.g. recently, a much larger effort to jointly analyze WES/WGS datasets derived from multiple centers/platforms/capture kits has been performed in the AD field (Holstege et al Nature Genetics 2022). This study includes, amongst others, the data presented in the current manuscript. How does the computational framework compare to the methods in Holstege et al? In that paper, specific attention was drawn to technical batch effects, e.g. due to use of restriction enzymes in certain capture kits, which may be affected by mutations in restriction sites. How are this and similar technical causes of batch effects addressed in the pipeline presented here?

Response: We thank the reviewer for the comment. The bioinformatics pipeline used in the current manuscript (which is compatible with CCDG/TOPMed) is different from what was used in Holstege *et al* Nature Genetics 2022. Those differences have been summarized in **Supplementary Table 12**. We have also included some text in the **DISCUSSION** for clarifications.

Some of the studies used in this manuscript were performed up to 10 years ago and were not published. Therefore, it is not possible for us to determine if the genomic DNA was

digested with enzymes, mechanically sheared or fragmented by transposomes prior to the exome capture step. We however did know that there are multiple biological steps in the WES sequencing library protocol that might create different technical biases.

Technical biases do exist in how these data were generated: different sequencing platforms, sequencing centers, and captures used (default vs custom). The variability of the captures can be explained by the portion of the genome the probes were designed to cover (**Table 3**) – two additional columns were added to reflect this design biases (first two in **Table 3**): the total number of regions included per capture at their default genome build, and the numbers after lift-over to GRCh38.

Therefore, to minimize the technical biases, in this current pipeline VCPA-WES we developed, we called all possible variants available in each sample, then perform quality control afterwards instead of limiting ourselves upfront at the variant calling step for each sample. This way joint genotyping will generate calls for variants at all sites prior to quality control even if that site is only present in one individual.

3. The methods section only reports the bioinformatics pipeline that was followed to process the WES data. It would be helpful to also include a description on how the results were analyzed, e.g. how was the Jaccard similarity measure computed?

Response: We thank the reviewer for the suggestion. We have extended in “Capture kit comparison” under **RESULTS** describing how the Jaccard similarity measure was computed.

4. The authors conclude that differences in sequencing platform or center did not affect processed CRAMs or 20x coverage. Was this conclusion based on eyeballing, or was this formally tested? The latter would be preferable, in which case the statistics used should also be included in the methods section.

Response: We thank the reviewer for the comments. We have included results from these statistical tests into “Data quality – WES CRAMs” under **RESULTS** and in **Supplementary Tables 1-10**.

We took the VCPA-WES generated CRAM metrics and compared those across different sequencing platforms or centers. We computed the decile values for each metric per sequencing platform/center, and compared every two distributions using Wilcoxon signed-rank test (pair-wise tests). Results were summarized below (statistically significant, Bonferroni corrected $p < 0.05$):

- Stratified by sequencing centers (**Figure 4a**), 11%, 3%, 8% and 76% per “percentage of mapped reads”, “percentage of duplicated reads”, “percentage of paired reads” and “quality of reads (based on a Q score of 30 [Q30])” respectively were statistically significant (Bonferroni corrected $p < 0.05$).
- Stratified by sequencing platforms (**Figure 4b**), 33%, 0%, 0% and 100% per “percentage of mapped reads”, “percentage of duplicated reads”, “percentage of

paired reads” and “quality of reads (based on a Q score of 30 [Q30])” respectively were statistically significant (Bonferroni corrected $p < 0.05$).

Sequencing platforms or centers have a bigger effect on “Quality of reads, Q30”, as this metric reflects the quality of the reads from the sequencers, which is directly dependent on the sequencing methods/protocols. The drastic differences observed for Q30 indicated that there are indeed differences at the sequencing methods level, yet after processing all the data using the same bioinformatics approach we proposed in this paper (VCPA-WES), the variabilities of data contributed by sequencing centers and platforms have much been reduced (as seen from the metrics at the mapping, duplicated or paired reads level).

We next performed the same analyses on 20X coverage instead of CRAM metrics, stratified by either the sequencing centers (**Figure 5a**), or the sequencing platforms (**Figure 5b**), all pair-wise tests (except one) were statistically significant (Bonferroni corrected $p < 0.05$). This is as expected as the 20X coverage was calculated on the capture regions per sample, of which the differences are due to the selection of various sequencing methods/protocols.

Reviewer #2 (Remarks to the Author):

This paper describes a joint calling strategy for the meta-analysis of Alzheimer’s Disease exomes, considering using capture kits and detecting batch effects during the QC process. The data has been deposited. Although the generation of genomic resources for the community is appreciated, joint calling and sample QC is now standardized through existing tools. This is especially true for a GATK-based pipeline and is usually presented as part of an analysis paper rather than a standalone paper. Examples of such datasets from the last five years include the gnomAD project, the Autism Sequencing Consortium, Deciphering Developmental Disorders, TOPMED, and UK10K studies, all of which handle multiple capture kits or include whole-genome and whole-exome sequencing data. Therefore, while this paper can still be valuable (as shared community resources are), a few areas need additional clarity and detail to maximize value for the scientific community.

Tables and Figures:

1. Table 1: Can you provide the number of cases and controls per study/capture to understand better how the study is distributed across captures?

Response: We thank the reviewer for this suggestion. We have added in **Table 1** the number of cases and controls per study/capture as requested. Here, the total number of samples were shown, with the proportion of cases displayed in percentages.

2. New figures: Because the study describes a modified calling pipeline, can a figure be added showing the step-by-step workflow of processing the data? Because the pipeline

largely uses the existing “VCPA” pipeline, it would help disentangle the genome and exome-specific components. I don’t think the details of the VCPA pipeline should be presumed.

Response: We thank the reviewer for this comment. We have included the step-by-step workflow of the pipeline into **Figure 1**. Those parts highlighted with a “star” are exome specific components (VCPA-WES).

3. Demographic information, especially related to genetic ancestry, should be reworded slightly in consideration

of <https://nap.nationalacademies.org/resource/26902/interactive/>.

More importantly, it would be helpful to have a PCA plot highlighting diversity within this sample. Separately, it would be helpful to see a plot showing the relatedness within this sample as an additional data set descriptor.

Response: We understand the reviewer’s concern and addressed through a detailed description of our population substructure assessment of the datasets and assignments of ancestry group labels to distinct subsets of the data. We refer specifically to the guidelines set forth by the NASEM report and we verified that our approach met the standards set forth. It should be noted that as these data have already been made available for public use, the ancestry group labels used reflect the data that have been released, and cannot practically be changed without a significant secondary release of the data, however the nomenclature used to identify these ancestry groups has been discussed and justified in accordance with the approach set forth in the report.

The description on how the ancestry subgroups were identified is described in “Quality control (QC) protocol for WES samples” under **METHODS**, and the suggested PCA plot is shown in **Figure 1 (RESULTS: “Population substructure analysis”)**.

Analysis:

4. From Figure 1, it appears that the overall overlap between captures is very low (0.586) when they should all target the exome. Can you provide more context as to why that is and how that affects the overall use of the data? Furthermore, what does this mean for capturing the coding region? Rather than focusing on the capture kit, we can look at the coverage of the coding region instead.

Response: Though the capture kits were in theory used for targeting exomes, some of the manufacturing companies have included miRNA or lncRNA sequences to the captures. This can be seen in an independent effort by UCSC, who tries to curate the diversified contents and file sizes of captures summarized here: “Exome Capture Probesets and Targeted Region”: https://genome.ucsc.edu/cgi-bin/hgTrackUi?g=exomeProbesets&hgsid=1066518897_QJL7hsBNGEhTnw6DgqcZaMG4YFB2.

To show the fundamental differences in designs across the captures, we included the original number of capture regions as well as the number of lifted-over capture regions

per captures (**Table 3**, new columns 1 and 2). As can be seen, one capture (Roche_SeqCap_EZ_Exome_Probes_v3.0_Target_Enrichment_Probes) has 1.5 times the number of probes compared to most of the others.

Coverage of the coding region (exons) as suggested by the reviewer is computed on 100 randomly selected samples (based on different studies-sequencing_center-capture combinations). Those were compared against the original 20x coverage calculated on the capture region in a scatter plot (**Supplementary Fig. 1**). The 20x coverage at the capture regions (88.5 ± 7.8) is similar to that of the coding regions (88.2 ± 2.5) across samples, though the 20x coverage values in coding regions are more uniform across samples. We have added these results in “Data quality – WES Compressed Reference-oriented Alignment Maps (CRAMs)” under **RESULTS**.

5. Is there a reason why the 20x coverage is used to determine the quality of these data?

Response: This is a standard/common practice to look at 20x coverage for WES. We have included in the text (under “Processing WES using VCPA at the individual sample level” in **METHODS**) multiple citations for referencing 20x coverage chosen for accessing the quality of the WES data:

Yu T.W., Chahrour M.H., Coulter M.E., Jiralerspong S., Okamura-Ikeda K., Ataman B., Schmitz-Abe K., Harmin D.A., Adli M., Malik A.N., D’Gama A.M., Lim E.T., Sanders S.J., Mochida G.H., Partlow J.N., Sunu C.M., Felie J.M., Rodriguez J., Nasir R.H., Ware J., Joseph R.M., Hill R.S., Kwan B.Y., Al-Saffar M., Mukaddes N.M., Hashmi A., Balkhy S., Gascon G.G., Hisama F.M., LeClair E., Poduri A., Oner O., Al-Saad S., Al-Awadi S.A., Bastaki L., Ben-Omran T., Teebi A., Al-Gazali L., Eapen V., Stevens C.R., Rappaport L., Gabriel S.B., Markianos K., State M.W., Greenberg M.E., Taniguchi H., Braverman N.E., Morrow E.M., & Walsh C.A. Using whole exome sequencing to identify inherited causes of autism. *Neuron* **77(2)**, 259-73 (2013).

Lelieveld S.H., Spielmann M., Mundlos S., Veltman J.A., & Gilissen C. Comparison of Exome and Genome Sequencing Technologies for the Complete Capture of Protein-Coding Regions. *Hum Mutat* **36(8)**:815-22 (2015).

Parla J.S., Iossifov I., Grabill I., Spector M.S., Kramer M., & McCombie W.R. A Comparative Analysis of Exome Capture. *Genome Biology* **12(9)**, R97 (2011).

6. Can you compare the distribution of coding variants (syn, mis, PTVs) in these data compared to other published data (like gnomAD)?

Response: We thank the reviewer for this suggestion. We have downloaded the gnomADv2.1 GRCh38 lifted over version (URL: <https://gnomad.broadinstitute.org/downloads#v2-liftover>) and performed the analyses.

In the released dataset, gnomADv2.1 (GRCh38) contains 17.2 million variants in 125,748 WES samples as compared to 8.2 million variants in 20,504 WES samples in the current dataset.

- 10.6 million (62%) and 4.0 million (49%) are coding variants in the gnomADv2.1 and the current WES data respectively.
- We next compared the percentage of variants that are annotated as ‘high impact’ in each set. There are 711,024 (6.7% of coding) and 220,987 (5.5% of coding) high impact coding variants in the gnomADv2.1 and the current dataset respectively.

We have added these into “Annotation results” under **RESULTS**.

7. As this document is a “Data Descriptor,” the overall text is light on analyses. Given the observation of batch effects (by capture), can you highlight how you would recommend that this data be analyzed for gene discovery? A useful “QC” metric compares the rates of synonymous variants between cases and controls on a gene-by-gene basis. This ensures that the QC is performed adequately. Can you provide this figure as an additional bit of analysis?

Response: As part of the documentation for data releases to ADSP, we typically include a “Quick Start Guide” that provides a detailed overview of recommended post-processing and analysis of the data. We have included this text in a new section in **METHODS** entitled “Recommended Post-QC Processing for Analysis” where we describe these steps, we hope that this adds more analysis-relevant detail so that the paper is no longer light on analysis.

In addition to this, in response to this and other concerns of the reviewers, we have added more extensive analyses of the data being shared to better characterize the resulting quality of the data from QC. We have taken the reviewers suggestion of comparing the rates of synonymous variants between cases and controls and added a new table summarizing these values exome-wide to the paper (**Table 4**). We have also included a figure to show the gene-by-gene synonymous variant frequency differences between cases and controls on studies/QC subsets that are of biggest sample size (**Figure 7**). We have added some text to in the “Data quality – variants” (**RESULTS**) describing what we observed here.

8. In that vein, the absolute number of coding variants (nhets, nhoms, nhomvar) in each cohort are useful metrics to see how comparable the sample is before and after QC. Can you provide that as a separate figure in the text (split by cohort, capture)?

Response: Based on the reviewer’s suggestions, we have constructed a table of REF/REF, REF/ALT, and ALT/ALT genotype counts summed across coding variants before and after QC for each QC subset (**Supplementary Table 11**). We have also shown in **Figure 8** the “Ratio of Post-QC to Pre-QC Genotype Counts” across QC subsets or all cohort-capture combinations. The ratio of all genotypes is fairly consistent across different QC subsets (0.922-0.956).

9. Code:

Usually, I do not over-emphasize code in a review of a paper, but because this is a resource paper, I took a deeper look. The code and the Gitbucket repo should be directly highlighted in the text (<https://bitbucket.org/NIAGADS/vcpa-pipeline/src/master/VCPA/>). From a look at the repo, most scripts are from an earlier iteration focusing on whole-genome sequence data (2018?). It will be helpful to highlight the workflow and scripts that are specific to the exome data and how to use them. The Supplementary Note includes two commands from GATK, and I would prefer that to be directly in the codebase. Furthermore, I cannot access the wiki, which might be helpful in understanding how to run the code.

Response: We thank the reviewer for checking this out. We have included in the text (**INTRODUCTION, DISCUSSION, CODE AVAILABILITY**) on where the code can be accessed. Public access is available for both the code repository (<https://bitbucket.org/NIAGADS/vcpa-pipeline/src/master/>) and wiki. A diagram summarizing the steps and scripts that are new in VCPA-WES pipeline have been highlighted in “stars” and included in **Figure 1**. VCPA-WES pipeline specific scripts were labeled with “_WES” in the file names for distinction purposes.

Minor:

10. There needs to be some editing and clarification of the acronyms in the text. A few examples include QC’ed and VCPA in the background.

Response: We thank the reviewer for pointing these out. We have fixed those in the background: “QC’ed” is for “quality controlled”. VCPA’s acronym was clarified in the manuscript and we did a bit of highlighting to make that clearer “**V**ariant **C**alling **P**ipeline and data management tool for **A**DS**P**”.

11. Can a variant-only file with allele frequencies like gnomAD be provided to the community?

Response: We thank the reviewer for asking this question. Unfortunately, due to NIH policy, we are not allowed to share the allele frequencies publicly. Users can obtain such data if they apply the data via DSS; the full VCF files with complete QC information is behind the firewall for users with approved applications. Research has shown that it is possible to use aggregated allele frequencies to infer back an individual’s identity (Homer et al., 2008; Liu et al., 2018). gnomAD is a database containing aggregated allele frequencies for subjects of multiple phenotypes, yet in our study, all subjects are of the same phenotype. The risk of leaking subjects’ info will directly infer that subject is part of an Alzheimer’s Disease related study and has a significant likelihood to be an Alzheimer’s disease patient.

References

- 1) Homer N., Szelinger S., Redman M., Duggan D., Tembe W., Muehling J., Pearson J.V., Stephan D.A., Nelson S.F., & Craig D.W. Resolving individuals

contributing trace amounts of DNA to highly complex mixtures using high-density SNP genotyping microarrays. *PLoS Genet* **4(8)**:e1000167 (2008).

- 2) Liu Y., Wan Z., Xia W., Kantarcioglu M., Vorobeychik Y., Clayton E.W., Kho A., Carrell D., Malin B.A. Detecting the Presence of an Individual in Phenotypic Summary Data. *AMIA Annu Symp Proc* **2018**:760-769 (2018).

Reviewer #3 (Remarks to the Author):

The Data Descriptor manuscript by Leung et al. describes an expanded cohort of whole exome sequencing data from over 20,000 individuals with a diagnosis of Alzheimers disease.

The manuscript is comprehensive, well written and describes an exceptional resource that is important for the research community.

I have a series of minor questions and comments about the manuscript that are focused on the improvement of the manuscript.

1. It would be informative to add the number of samples in the study that used each capture technologies to table 3.

Response: We thank the reviewer for the suggestion. We have included in **Table 1** the number of samples in each study broken down by capture technologies.

2. For the comparison of 20x sequence coverage (Figure 3 and text), it is unclear filtering of reads if any was performed prior to assessment (ie were duplicate reads, unpaired reads and/or >Q30 mapping score reads excluded).

Response: We thank the reviewer for the comment. When the 20x sequence coverage was compared across sequencing centers and platforms, no bad quality reads were filtered prior to assessment. We have clarified these in our text in "Processing WES using VCPA at the individual sample level" under **METHODS**.

3. One area which is unclear in the current manuscript is the quality of genotype calls across sites. I think the manuscript would benefit from an assessment per variant site of the genotype call rate when genotype calls with $GQ < 20$ (or some justifiable threshold) are set to missing - a histogram across variant sites would be informative (potentially also stratified by capture platform). As this will really get to the crux of the challenge with this call set, which is that due to the differences in capture technologies one would not expect high quality data for all samples at all sites.

Response: We thank the reviewer for the suggestion. We have generated histograms by subset showing decile bins containing exome-wide counts of variants with the range of call rates shown, with two call rates depicted: one including genotypes that failed our genotype-level QC criterion of having $DP < 10$ and/or $GQ < 20$, and one-excluding them. These plots have been included into **Supplementary Figure 2**, with a description added

to “Data quality – variants” of the **RESULTS** section. As expected the average call rates show modest improvement with the exclusion of low-quality genotypes across all variants across QC subsets.

4. The statement in the discussion that the dataset is 'free of batch effects' is an overstatement and I am sure that it would be possible to identify batch effect by call-rate (see above comment), the authors may want to revise this statement to reflect that they have employed a harmonisation approach to alignment and variant calling to minimise the impact of different analytical pipelines.

Response: We thank the reviewer for pointing this out. We have used the wordings as suggested by the reviewer in the **DISCUSSION** session.

REVIEWERS' COMMENTS

Reviewer #1 (Remarks to the Author):

The authors have addressed my comments well

Reviewer #3 (Remarks to the Author):

The authors have adequately addressed my comments and the manuscript is improved from the original version.

Reviewer #4 (Remarks to the Author):

Most comments from previous Reviewer #2 have been thoroughly taken into accounts. Some elements require further clarifications though. This is mostly to make sure the readers understand correctly the pieces of information that have been added to the manuscript at the review stage.

See attached pdf for detailed comments on responses to Reviewer 2.

Two additional remarks:

- OPRL1 and GAS2L2 are cited as "two novel genes" in INTRODUCTION (I.79).

This assertion is wrong. Not only the replication does not support this association signal in the discovery paper itself, but the association was not found at all in the Holstege et al paper later on, despite higher statistical power. These genes are most probably false positive signals.

- it is well known that at equal sequencing depth, heterozygous genotypes and homozygous genotypes, do not have the same chance of reaching good genotype quality. The pipeline sets low quality genotypes to ./.. This strategy will result in a strong imbalance of true 0/0 and 0/1 set to ./.. because low quality, particularly at low sequencing depths. It seems that a paper describing how to best analyse WES data with diverse capture kits and diverse sequencing depths should adress these issues when describing the best post QC strategy.

Reviewer #2 (Remarks to the Author):

This paper describes a joint calling strategy for the meta-analysis of Alzheimer's Disease exomes, considering using capture kits and detecting batch effects during the QC process. The data has been deposited. Although the generation of genomic resources for the community is appreciated, joint calling and sample QC is now standardized through existing tools. This is especially true for a GATK-based pipeline **and is usually presented as part of an analysis paper rather than a standalone paper.**

Examples of such datasets from the last five years include the gnomAD project, the Autism Sequencing Consortium, Deciphering Developmental Disorders, TOPMED, and UK10K studies, all of which handle multiple capture kits or include whole-genome and whole-exome sequencing data. Therefore, while this paper can still be valuable (as shared community resources are), a few areas need additional clarity and detail to maximize value for the scientific community.

Tables and Figures:

1. Table 1: Can you provide the number of cases and controls per study/capture to understand better how the study is distributed across captures?

Response: We thank the reviewer for this suggestion. We have added in Table 1 the number of cases and controls per study/capture as requested. Here, the total number of samples were shown, with the proportion of cases displayed in percentages.

Reviewer response: This table is very useful indeed to understand the dataset structure. However, I am really sorry, but I do not understand what the denominator is for the proportion of cases displayed in percentages. All percentages are very close to 100% and therefore do not sum up to 100% either in line or column, and they are too often close to 100% to represent the % of cases among the sequenced samples within a given study/capture combination. This should be clarified.

2. New figures: Because the study describes a modified calling pipeline, can a figure be added showing the step-by-step workflow of processing the data? Because the pipeline largely uses the existing "VCPA" pipeline, it would help disentangle the genome and exome-specific components. I don't think the details of the VCPA pipeline should be presumed.

Response: We thank the reviewer for this comment. We have included the step-by-step workflow of the pipeline into Figure 1. Those parts highlighted with a "star" are exome specific components (VCPA-WES).

Reviewer response: OK.

3. Demographic information, especially related to genetic ancestry, should be reworded slightly in consideration of <https://nap.nationalacademies.org/resource/26902/interactive/>.

More importantly, it would be helpful to have a PCA plot highlighting diversity within this sample. Separately, it would be helpful to see a plot showing the relatedness within this sample as an additional data set descriptor.

Response: We understand the reviewer's concern and addressed through a detailed description of our population substructure assessment of the datasets and assignments of ancestry group labels to distinct subsets of the data. We refer specifically to the guidelines set forth by the NASEM report and we verified that our approach met the standards set forth. It should be noted that as these data have already been made available for public use, the ancestry group labels used reflect the data that have been released, and cannot practically be changed without a significant secondary release of the data, however the nomenclature used to identify these ancestry groups has been discussed and justified in accordance with the approach set forth in the report.

The description on how the ancestry subgroups were identified is described in "Quality control (QC) protocol for WES samples" under METHODS, and the suggested PCA plot is shown in Figure 1 (RESULTS: "Population substructure analysis").

Reviewer response: Thank you for these added explanations. The legend of Figure 1, panels a to c, is difficult to understand because it seems that for panels b and c, the targeted population is represented in black dots, but for panel a, EUR samples are not in black. Instead CHB samples are highlighted, which is confusing. Is it possible to keep the targeted population as black highlight for each panel?

Besides, on panel b, could you explain why some samples that clearly colocalize with JPT or CHB populations (+ two other very close) are included in the AA study? Do they match the +/-3SD rule? If so, is this rule too lenient?

From the Figures, it is hard to understand whether the ancestry analysis defines a partition of non-overlapping groups, or if these ancestry groups are potentially embedded within each other. Could you clarify this point?

Analysis:

4. From Figure 1, it appears that the overall overlap between captures is very low (0.586) when they should all target the exome. Can you provide more context as to why that is and how that affects the overall use of the data? Furthermore, what does this mean for capturing the coding region? Rather than focusing on the capture kit, we can look at the coverage of the coding region instead.

Response: Though the capture kits were in theory used for targeting exomes, some of the manufacturing companies have included miRNA or lncRNA sequences to the captures. This can be seen in an independent effort by UCSC, who tries to curate the diversified contents and file sizes of captures summarized here: "Exome Capture Probesets and Targeted Region": https://genome.ucsc.edu/cgi-bin/hgTrackUi?g=exomeProbesets&hgsid=1066518897_QJL7hsBNGEhTnw6DgqcZaM G4YFB2 .

To show the fundamental differences in designs across the captures, we included the original number of capture regions as well as the number of lifted-over capture regions per captures (Table 3, new columns 1 and 2). As can be seen, one capture (Roche_SeqCap_EZ_Exome_Probes_v3.0_Target_Enrichment_Probes) has 1.5 times the number of probes compared to most of the others.

Coverage of the coding region (exons) as suggested by the reviewer is computed on 100 randomly selected samples (based on different studies-sequencing_center-capture combinations). Those were compared against the original 20x coverage calculated on the capture region in a scatter plot (Supplementary Fig. 1). The 20x coverage at the capture regions (88.5 ± 7.8) is similar to that of the coding regions (88.2 ± 2.5) across samples, though the 20x coverage values in coding regions are more uniform across samples. We have added these results in “Data quality – WES Compressed Reference-oriented Alignment Maps (CRAMs)” under RESULTS.

Reviewer response: ok, this is very helpful.

5. Is there a reason why the 20x coverage is used to determine the quality of these data?

Response: This is a standard/common practice to look at 20x coverage for WES. We have included in the text (under “Processing WES using VCPA at the individual sample level” in METHODS) multiple citations for referencing 20x coverage chosen for accessing the quality of the WES data:

Yu T.W., Chahrour M.H., Coulter M.E., Jiralerspong S., Okamura-Ikeda K., Ataman B., Schmitz-Abe K., Harmin D.A., Adli M., Malik A.N., D’Gama A.M., Lim E.T., Sanders S.J., Mochida G.H., Partlow J.N., Sunu C.M., Felie J.M., Rodriguez J., Nasir R.H., Ware J., Joseph R.M., Hill R.S., Kwan B.Y., Al-Saffar M., Mukaddes N.M., Hashmi A., Balkhy S., Gascon G.G., Hisama F.M., LeClair E., Poduri A., Oner O., Al-Saad S., Al-Awadi S.A., Bastaki L., Ben-Omran T., Teebi A., Al-Gazali L., Eapen V., Stevens C.R., Rappaport L., Gabriel S.B., Markianos K., State M.W., Greenberg M.E., Taniguchi H., Braverman N.E., Morrow E.M., & Walsh C.A. Using whole exome sequencing to identify inherited causes of autism. *Neuron* 77(2), 259-73 (2013).

Lelieveld S.H., Spielmann M., Mundlos S., Veltman J.A., & Gilissen C. Comparison of Exome and Genome Sequencing Technologies for the Complete Capture of Protein-Coding Regions. *Hum Mutat* 36(8):815-22 (2015).

Parla J.S., Iossifov I., Grabill I., Spector M.S., Kramer M., & McCombie W.R. A Comparative Analysis of Exome Capture. *Genome Biology* 12(9), R97 (2011).

Reviewer response: ok, although all those references are getting old. With new sequencing standards and lower prices, the depth of sequencing is a lot higher now than what it was in 2011. 20x is still rather low to call with confidence ultra-rare variants or singletons. It could be noisy if the allelic balance is not of a perfect 50%. It is possible that asking for 30x (or more) would be more informative and more discriminant across platforms and capture kits.

6. Can you compare the distribution of coding variants (syn, mis, PTVs) in these data compared to other published data (like gnomAD)?

Response: We thank the reviewer for this suggestion. We have downloaded the gnomADv2.1 GRCh38 lifted over version (URL: <https://gnomad.broadinstitute.org/downloads#v2-liftover>) and performed the analyses.

In the released dataset, gnomADv2.1 (GRCh38) contains 17.2 million variants in 125,748 WES samples as compared to 8.2 million variants in 20,504 WES samples in the current dataset.

- - 10.6 million (62%) and 4.0 million (49%) are coding variants in the gnomADv2.1 and the current WES data respectively.
- - We next compared the percentage of variants that are annotated as 'high impact' in each set. There are 711,024 (6.7% of coding) and 220,987 (5.5% of coding) high impact coding variants in the gnomADv2.1 and the current dataset respectively.

We have added these into "Annotation results" under RESULTS.

Reviewer response: ok, this answers the comment.

7. As this document is a "Data Descriptor," the overall text is light on analyses. Given the observation of batch effects (by capture), can you highlight how you would recommend that this data be analyzed for gene discovery? A useful "QC" metric compares the rates of synonymous variants between cases and controls on a gene-by-gene basis. This ensures that the QC is performed adequately. Can you provide this figure as an additional bit of analysis?

Response: As part of the documentation for data releases to ADSP, we typically include a "Quick Start Guide" that provides a detailed overview of recommended post-processing and analysis of the data. We have included this text in a new section in METHODS entitled "Recommended Post-QC Processing for Analysis" where we describe these steps, we hope that this adds more analysis-relevant detail so that the paper is no longer light on analysis.

In addition to this, in response to this and other concerns of the reviewers, we have added more extensive analyses of the data being shared to better characterize the resulting quality of the data from QC. We have taken the reviewers suggestion of comparing the rates of synonymous variants between cases and controls and added a new table summarizing these values exome-wide to the paper (Table 4). We have also included a figure to show the gene-by-gene synonymous variant frequency differences between cases and controls on studies/QC subsets that are of biggest sample size (Figure 7). We have added some text to in the "Data quality – variants" (RESULTS) describing what we observed here.

Reviewer response: ok, this answers the comment.

8. In that vein, the absolute number of coding variants (nhets, nhoms, nhomvar) in each cohort are useful metrics to see how comparable the sample is before and after QC. Can you provide that as a separate figure in the text (split by cohort, capture)?

Response: Based on the reviewer's suggestions, we have constructed a table of REF/REF, REF/ALT, and ALT/ALT genotype counts summed across coding variants before and after QC for each QC subset (Supplementary Table 11). We have also shown in Figure 8 the "Ratio of Post-QC to Pre-QC Genotype Counts" across QC subsets or all cohort-capture combinations. The ratio of all genotypes is fairly consistent across different QC subsets (0.922-0.956).

Reviewer response: ok, this answers the comment.

9. Code:

Usually, I do not over-emphasize code in a review of a paper, but because this is a resource paper, I took a deeper look. The code and the Gitbucket repo should be directly highlighted in the text (<https://bitbucket.org/NIAGADS/vcpa-pipeline/src/master/VCPA/>). From a look at the repo, most scripts are from an earlier iteration focusing on whole-genome sequence data (2018?). It will be helpful to highlight the workflow and scripts that are specific to the exome data and how to use them. The Supplementary Note includes two commands from GATK, and I would prefer that to be directly in the codebase. Furthermore, I cannot access the wiki, which might be helpful in understanding how to run the code.

Response: We thank the reviewer for checking this out. We have included in the text (INTRODUCTION, DISCUSSION, CODE AVAILABILITY) on where the code can be accessed. Public access is available for both the code repository (<https://bitbucket.org/NIAGADS/vcpa-pipeline/src/master/>) and wiki. A diagram summarizing the steps and scripts that are new in VCPA-WES pipeline have been highlighted in "stars" and included in Figure 1. VCPA-WES pipeline specific scripts were labeled with "_WES" in the file names for distinction purposes.

Reviewer response: ok, this answers the comment.

11. Can a variant-only file with allele frequencies like gnomAD be provided to the community?

Response: We thank the reviewer for asking this question. Unfortunately, due to NIH policy, we are not allowed to share the allele frequencies publicly. Users can obtain such data if they apply the data via DSS; the full VCF files with complete QC information is behind the firewall for users with approved applications. Research has shown that it is possible to use aggregated allele frequencies to infer back an individual's identity (Homer et al., 2008; Liu et al., 2018). gnomAD is a database containing aggregated allele frequencies for subjects of multiple phenotypes, yet in our study, all subjects are of the same phenotype. The risk of leaking subjects' info will directly infer that subject is part of an Alzheimer's Disease related study and has a significant likelihood to be an Alzheimer's disease patient.

Reviewer response: ok, this answers the comment.

References

1) Homer N., Szelinger S., Redman M., Duggan D., Tembe W., Muehling J.,

Pearson J.V., Stephan D.A., Nelson S.F., & Craig D.W. Resolving individuals contributing trace amounts of DNA to highly complex mixtures using high-density SNP genotyping microarrays. *PLoS Genet* 4(8):e1000167 (2008).

2) Liu Y., Wan Z., Xia W., Kantarcioglu M., Vorobeychik Y., Clayton E.W., Kho A., Carrell D., Malin B.A. Detecting the Presence of an Individual in Phenotypic Summary Data. *AMIA Annu Symp Proc* 2018:760-769 (2018).

Reviewer #4

We thank the reviewer for going through the previous edits we made to address Reviewer #2's questions and also provide additional comments to improve the quality of the manuscript. In the following, we extracted the questions (and thread) from which the Reviewer #4 has comments on. In the following, the black texts are from Reviewer #2, blue texts are from Reviewer #4, and our responses in red.

1) Table 1: Can you provide the number of cases and controls per study/capture to understand better how the study is distributed across captures?

Reviewer response: This table is very useful indeed to understand the dataset structure. However, I am really sorry, but I do not understand what the denominator is for the proportion of cases displayed in percentages. All percentages are very close to 100% and therefore do not sum up to 100% either in line or column, and they are too often close to 100% to represent the % of cases among the sequenced samples within a given study/capture combination. This should be clarified.

Response: We thank the reviewer for the comment. We found a bug in generating these numbers and have updated the percentages in the table, yet the explanation of these numbers remains unchanged. Take the first column as an example. There are 61 samples sequenced using the "Agilent_WES_v3_capture_region" kit in the "Miami_Families" study, 74% of the 61 samples are cases (i.e. 45 samples). There are 2 studies indeed only have cases sequenced (CBD and PSP), that's why the percentages shown are 100%.

3) Demographic information especially related to genetic ancestry, should be reworded slightly in consideration of <https://nap.nationalacademies.org/resource/26902/interactive/>. More importantly, it would be helpful to have a PCA plot highlighting diversity within this sample. Separately, it would be helpful to see a plot showing the relatedness within this sample as an additional data set descriptor.

Reviewer response: Thank you for these added explanations. The legend of Figure 1, panels a to c, is difficult to understand because it seems that for panels b and c, the targeted population is represented in black dots, but for panel a, EUR samples are not in black. Instead CHB samples are highlighted, which is confusing. Is it possible to keep the targeted population as black highlight for each panel?

Response: Thank you for your kind suggestions. We have revised the plots to have consistent color designation and improved visualization across all panels, including selecting darker colors for previously lightly visible symbols and using smaller black dots to represent each ADSP ancestry sub-group on each plot to allow the clustering of overlapping 1000 Genomes reference sample sets to be more clearly visible as well.

Besides, on panel b, could you explain why some samples that clearly colocalize with JPT or CHB populations (+ two other very close) are included in the AA study? Do they match the $\pm 3SD$ rule? If so, is this rule too lenient?

Response: The outliers are present among the samples depicted, although we did not visually identify which samples were excluded here as the boundary of the $\pm 3SD$ was not easily visualized in the plot, however, we now have noted how many of the samples were excluded as outliers for each ancestry group in the figure caption text.

From the Figures, it is hard to understand whether the ancestry analysis defines a partition of non-overlapping groups, or if these ancestry groups are potentially embedded within each other. Could you clarify this point?

Response: The ancestry analysis depicted in the Figure 2 characterizes the approximate ancestral membership of the WES 20k samples separated into three non-overlapping groups initially identified by self-reported race/ethnicity. Although race/ethnicity are poor proxies for genetic ancestry, these categorizations may in large part capture specific ancestral and admixed ancestry backgrounds; in this instance, these three overlapping groups were determined on the basis of source study ascertainment designs, which were predominantly based on self-identified race/ethnicity (e.g., recruitment of “African American” participants). These categorizations were mostly co-terminus with individual source datasets, capturing sequencing center-, sequencing platform-, and capture-kit specific batch effects. Because including individuals showing ancestry outlier status with respect to their original race/ethnicity categorization would have led to small cell sizes when performing analytical adjustment for sequencing center, sequencing platform, and capture kit effects, a small number of individuals whose genetic ancestry did not overlap with others in their race/ethnicity categorization (32 samples in NHW; 29 samples in AA; 0 samples in HIS) were suggested/flagged to be dropped for within-ancestry analysis.

To clarify this, we have modified the text under “Quality Control (QC) Protocol for WES Samples” to reflect how samples were categorized.

5) Is there a reason why the 20x coverage is used to determine the quality of these data?

Reviewer response: ok, although all those references are getting old. With new sequencing standards and lower prices, the depth of sequencing is a lot higher now than what it was in 2011. 20x is still rather low to call with confidence ultra-rare variants or singletons. It could be noisy if the allelic balance is not of a perfect 50%. It is possible that asking for 30x (or more) would be more informative and more discriminant across platforms and capture kits.

Response: We thank the reviewer for the kind suggestion. We agree that 30x is the norm for checking coverage for whole genome sequencing but may not be the case for WES data for two reasons. First, the WES data were generated in different labs across different times (2010-2021); most of the data were generated using older technologies

and would expect to have a lower coverage. Second, recently, gnomAD v.4 released their data which contains 730,947 exomes. They are also using 20x to consider if an exome calling interval is high coverage or not (see <https://gnomad.broadinstitute.org/news/2023-11-gnomad-v4-0/>). We included two more recent references to justify the use of 20x in most exome studies (LaDuca 2017, PMID: 28152038, Lee 2021, PMID: 34016036).

The follow two are the additional comments from Reviewer #4

A) OPRL1 and GAS2L2 are cited as “two novel genes” in INTRODUCTION (1.79). This assertion is wrong. Not only the replication does not support his association signal in the discovery paper itself, but the association was not found at all in the Holstege et al paper later on, despite higher statistical power. These genes are most probably false positive signals.

Response: We thank the reviewer for pointing this out. The three novel genes found in Bis *et al* 2020 should be *IGHG3*, *STAG3* and *ZNF655*. These results remained significant after multiple test correction and were confirmed in or strengthened by a replication sample comprised of four independent datasets in this study. We have updated in the INTRODUCTION to reflect this change.

B) It is well known that at equal sequencing depth, heterozygous genotypes and homozygous genotypes, do not have the same chance of reaching good genotype quality. The pipeline sets low quality genotypes to ‘./.’. This strategy will result in a strong imbalance of true ‘0/0’ and ‘0/1’ set to ‘./.’ because [of] low quality, particularly at low sequencing depths. It seems that a paper describing how to best analyze WES data with diverse capture kits and diverse sequencing depths should address these issues when describing the best post QC strategy.

Response: We appreciate the importance of the reviewer’s comment; if available, we welcome any specific literature or references the reviewer may recommend us to refer to in the manuscript to address this concern, as we have not been able to identify an ideal reference to refer to in our modification of the text (due diligence prevents us from saying that this imbalance is ‘well-known’ in the text of the manuscript).

Because of the broad spectrum of sources for potential variant and genotype quality issues, including differences in sequencing center, sequencing platform, and capture-kit, among others, it was not practical to address all quality issues with separate/independent strategies without potentially inducing additional biases. As a result, our QC strategy (a) was conservative in identifying genotypes and variants of potentially low quality and (b) allowed us to retain information to allow recalling of

missing genotypes for interested end-users of the data. Briefly, by automatically flagging variants for exclusion that fell outside of genomic regions targeted by capture kits on a per-capture kit basis, we systematically identify the majority of variants with low average read depths susceptible to this bias. This filter combined with the variant-level GATK “PASS” filter excluded all but 3.5% of genotypes that were set to missing due to low DP/low GQ, as most remaining variants had higher average call rates. Additional QC filters reduced the number of variants containing genotype set to missing even further. We believe our results show that the effects of this bias were largely mitigated using our overly conservative QC filtering approach to allow for analysis of variants of consistently high quality across source studies, sequencing center/assay differences, and capture kits.

We briefly addressed this with an additional sentence at the end of the section “Recommended Post-QC Processing for Analysis.”